



# Measurement report: Characterization of the vertical distribution of airborne *Pinus* pollen in the atmosphere with lidar-derived profiles: a modelling case study in the region of Barcelona, NE Spain

Michaël Sicard[1,2], Oriol Jorba[3], Jiang Ji Ho[4], Rebeca Izquierdo[4,5,6], Concepción De Linares[5,7], Marta Alarcón[4], Adolfo Comerón[1], Jordina Belmonte[5,6]

[1]CommSensLab, Dept. of Signal Theory and Communications, Universitat Politècnica de Catalunya, 08034, Barcelona, Spain
[2]Ciències i Tecnologies de l'Espai - Centre de Recerca de l'Aeronàutica i de l'Espai / Institut d'Estudis Espacials de Catalunya (CTE-CRAE / IEEC), Universitat Politècnica de Catalunya, 08034, Barcelona, Spain
[3]Barcelona Supercomputing Center, 08034, Barcelona, Spain
[4]Departament de Física, Universitat Politècnica de Catalunya, 08019, Barcelona, Spain
[5]Departament de Biologia Animal, Biologia Vegetal i Ecologia, Universitat Autònoma de Barcelona, 08193, Bellaterra, Spain
[6]Institut de Ciencia i Tecnología Ambientals (ICTA), Universitat Autònoma de Barcelona, 08193, Bellaterra, Spain
[7]Department of Botany, University of Granada, 18071, Granada, Spain

*Correspondence to*: Michaël Sicard (msicard@tsc.upc.edu)

**Abstract.** This paper investigates the mechanisms involved in the dispersion, structure and mixing in the vertical column of atmospheric pollen. The methodology used employs observations of pollen concentration obtained from Hirst samplers (we will refer to as surface pollen) and vertical distribution (polarization-sensitive lidar) as well as nested numerical simulations with an atmospheric transport model and a simplified pollen module developed especially for this study. The study focuses on

the predominant pollen type, *Pinus*, of the intense pollination event which occurred in the region of Barcelona, Catalonia, NE Spain, during 27 – 31 March, 2015. First, conversion formulas are expressed to convert lidar-derived total backscatter coefficient and model-derived mass concentration into pollen grains concentration, the magnitude measured at the surface by means of aerobiological methods, and for the first time ever, a relationship between optical and mass properties of atmospheric pollen, through the estimation of the so-called specific extinction cross-section, is quantified in ambient conditions. Second, the model horizontal representativeness is assessed through comparison between nested pollen simulations at 9, 3 and 1 km

horizontal resolution and observed meteorological and aerobiological variables at seven sites around Catalonia. Finally, hourly observations of surface and column concentration in Barcelona are analysed with the different numerical simulations at increasing horizontal resolution and varying sedimentation/deposition parameters. We find that the 9 or 3 km simulations are less sensitive to the meteorology errors hence they should be preferred for specific forecasting applications. The largest

discrepancies between measured surface (Hirst) and column (lidar) concentrations occur during nighttime: only residual pollen is detected in the column whereas it is present at the surface. The main reason is related to the lidar characteristics which has a lowest useful range bin at ~225 m, above the usually very thin nocturnal stable boundary layer. At the hour of the day of maximum insolation, the pollen layer does not extend up to the top of the planetary boundary layer according to the observations (lidar), probably because of gravity effects; however, the model simulates the pollen plume up to the top of the

planetary boundary layer, resulting in an overestimation of the pollen load. Besides the large size and weight of *Pinus* grains, sedimentation/deposition processes have only a limited impact on the model vertical concentration in contrast to the emission processes. For further modelling research, emphasis is put on the accurate knowledge of plant/tree spatial distribution, density and type, as well as on the establishment of reliable phenology functions.



## 1. Introduction

Pollen is a very important biological structure present all over the world. It functions as a container in which the male gametophyte generation of the angiosperms and gymnosperms is housed (Moore and Weeb, 1983, Skjøth et al., 2013) and is responsible of the gene flow (Burczyk et al., 2004; Ellstrand, 1992; Ennos, 1994). To be functional, mature pollen must be transported from the place where it is generated to the female structures of a flower of the same species, a process named pollination. Several pollination types exist, one of them being anemophily. Anemophily occurs when pollen grains are passively transported by the air. In this case, pollen behaves as biogenic aerosol and constitutes a substantial fraction of the mass of particulate matter in the air during the flowering season. Consequently, pollen can have strong health effects, causing allergenic rhinitis and asthma (D'Amato et al., 2007; Sofiev et al., 2017). The study of the pollen transport in the atmosphere is a relevant topic, not only because it allows evaluating the potential risks for human health and preventing its effects, but also because it will make possible a better comprehension of the spatial distribution of the species (Belmonte et al., 2008; Schmidt-Lebuhn et al., 2007; Sharma and Kanduri, 2007; Smouse et al., 2001).

*Pinus* is a dominant genus in the forested areas of the Northern Hemisphere and this is also true in the region of Barcelona (Catalonia, NE Spain), the region of the study of this work. Following the Catalan Ecological and Forestry Inventory (CEFI, Gracia et al., 2000-2004), 61% (2 million ha) of the total surface of Catalonia is covered by forests and *Pinus* (*P. halepensis*, *P. nigra*, *P. pinaster*, *P. pinea*, *P. sylvestris* and *P. uncinata*) accounts for 39.2 % of this area. Because of the abundance of pines in the territory, their huge pollen production and the pollen dispersion by anemophily, *Pinus* is one of the most abundant taxa in the atmospheric pollen spectra (Belmonte and Roure, 1991). *Pinus* pollen is in the top 3 of the most abundant pollen types in almost all stations of the Catalan Aerobiological Network (Xarxa Aerobiològica de Catalunya, XAC, https://lap.uab.cat/aerobiologia/en/). Although not considered an allergenic species, its abundance in the region makes this pollen type of interest to understand the atmospheric transport processes affecting biological aerosols.

In the last decades, several works have studied the transport of pollen species in the atmosphere developing different numerical models and comparing the results with in-situ surface observations. Such models include a source term and a dispersion module. The source term characterizes the pollen emission considering the start, end and duration of the pollen season (e.g., Sofiev et al., 2013) and the diurnal profiles of the emission fluxes (e.g., Helbig et al., 2004; Sofiev et al. 2013) with parameterizations derived from statistical analysis of available observations (surface pollen counts and meteorological variables). The seminal works of Helbig et al. (2004) and Sofiev et al. (2006) showed the value of such models to study the transport of pollen and their application as forecasting tools. Different models have been developed since then to model the transport of birch (Sofiev et al. 2006; Vogel et al., 2008; Efstathiou et al., 2011; Zink et al. 2013; Zhang et al., 2014), ragweed (Efstathiou et al., 2011; Zink et al., 2013; Prank et al., 2013; Wozniak and Steiner, 2017), grass (Zhang et al., 2014; Wozniak and Steiner, 2017), olive (Zhang et al., 2014; Sofiev et al., 2017), broadleaf tree pollen (Helbig et al., 2004; Wozniak and Steiner, 2017), and evergreen needleleaf tree pollen (Wozniak and Steiner, 2017) over regional domains with horizontal resolutions ranging from 50 to 10 km. Although advancements in the field have been achieved, current models still present significant limitations to reproduce the life cycle of pollen as highlighted in the model intercomparison works of Sofiev et al. (2015, 2017), where different transport models were used to study the pollination season of birch and olive with large variability among them. Nowadays, the Copernicus Atmosphere Monitoring Service (CAMS) Regional Production provides forecasts at the European continental scale of pollen concentration for birch, olive, grasses, ragweed, and alder using a multi-model ensemble approach (https://www.regional.atmosphere.copernicus.eu). While the evaluation of these simulations is almost always performed against in-situ pollen concentration measurements, nearly no information is known about their performance in the atmospheric column.

Scattering coupled to depolarizing properties, on the one hand, and emission of fluorescence spectra when excited with UV radiation of some chemical substances contained in pollen and bioparticles in general, on the other hand, are the two main properties of pollen grains allowing their remote detection in the atmospheric column with lidar techniques. The first property



makes elastic, polarization-sensitive lidar systems powerful instruments for the detection of atmospheric pollen. The number of articles from the lidar community dealing with this topic has increased in recent years (Sassen, 2008; Noh et al., 2013; Sicard et al., 2016a; Bohlmann et al., 2019; Shang et al., 2020). The second property of pollen to emit characteristic fluorescence spectra is known since a few years now (Sugimoto et al., 2012; Sharma et al., 2015; Wojtanowski et al., 2015;

Rao et al., 2017; Richardson et al., 2019) and can be detected by the technique of the so-called laser induced fluorescence lidars. Some authors even detected fluorescence effects of aerosol of biogenic origin with the water vapor channel of an elastic/Raman lidar system (Immler et al., 2005). Although the use of lidar-derived, range-resolved information on the optical properties of pollen is available, it has never been used in a generalized manner for validating modelling experiments trying to reproduce the pollen release and transport in the atmosphere, neither has it been used as a complementary tool to understand

the pollen vertical dispersion, distribution and mixing. Some tentative exercises, published in reviewed proceedings of international conferences, using modelling to help understand the observed vertical distribution of *Pinus* and *Platanus* pollen from lidar observations in the region of Barcelona have been conducted by Sicard et al. (2016b, 2017, 2019). The present journal paper is the apogee of our knowledge acquired throughout a continuous effort since 2016.

The objective of the paper is to improve our understanding of pollen vertical distribution in the atmosphere by combining in-
situ concentration (Hirst), columnar optical property (lidar) measurements and dispersion modelling. The paper is focused on the mechanisms responsible of the pollen vertical dispersion and mixing and how the pollen vertical structure impacts its horizontal transport. For that, the pollen type selected is *Pinus*, one of the most abundant (third position in the spectra of the period 1994-2020, after the ornamental species *Platanus* and Cupressaceae) in the region of Barcelona, our region of study. This work presents nested numerical simulations up to 1 km horizontal resolution of the dispersion of *Pinus* pollen in the

atmosphere which occurred during a 5-day pollination event in the region of Barcelona during 27 – 31 March, 2015 (Sicard et al., 2016a). The model evaluation in the atmospheric column is performed against continuous lidar measurements conducted in Barcelona city. The results are discussed in terms of the pollen parametrization and the schemes used in the model. The present journal paper is the first one of its kind studying the vertical structure of a pollen species by means of nested numerical simulations.

## 25 **2. Instrumentation**

Pollen grain daily concentration was measured by the Aerobiological Network of Catalonia (Xarxa Aerobiològica de Catalunya, XAC) at 7 sites around Catalonia (NE Spain). At the Barcelona site (2.165º E, 41.394° N, 67 m a.s.l.) the pollen concentration was also measured at a time resolution of an hour. The meteorology was taken from the 7 nearest stations to the XAC sites of the Automatic Weather Stations Network (Xarxa d'Estacions Meteorològiques Automàtiques, XEMA,

http://en.meteocat.gencat.cat/xema) of the Meteorological Service of Catalonia, Meteocat. The coordinates of these stations are reported in Table 1 and their location is shown in Figure 1a. Hourly profiles of particle backscatter coefficient and linear volume and particle depolarization ratios were acquired at the Remote Sensing Lab (RSLab) in the North Campus of the Universitat Politècnica de Catalunya (2.112º E, 41.389º N, 115 m a.s.l.), approximately 4.4 km to the west of the Barcelona XAC site.

## 35 **2.1. Airborne pollen sampling**

Pollen samples are obtained using volumetric suction pollen trap based on the impact principle (Hirst, 1952), the standardized method in European aerobiological networks (Galán et al., 2014). The Hirst sampler (Hirst, 1952) is calibrated to handle a flow of 10 litre of air per minute, thus matching the human breathing rate. Pollen grains are impacted on a cylindrical drum covered by a Melinex film coated with silicon fluid (LANZONI s.r.l. ®) as trapping surface. The drum rotates at 2 mm per

hour; so each 48 mm represents 24 hours of continuous sampling. The drum is changed weekly and the exposed tape is cut





into pieces, each one corresponding to one day. Pollen grains are counted under light microscope, at 600X magnification. Daily average pollen counts are obtained following the standardized Spanish method (Galán et al., 2007), consisting in running four longitudinal sweeps along the 24-h slide for daily data, identifying and counting each pollen type found. To obtain the hourly concentrations, twenty-four continuous transversal sweeps separated every 2 mm along the daily-sample slide are

analyzed, since the drum rotates at a speed of 2 mm per hour. Daily and intra-diurnal (hourly) pollen concentrations are obtained converting the pollen counts into particles per cubic meter of air, taking into account the proportion of the sample surface analyzed and the air intake of the Hirst pollen trap (10 l min$^{-1}$).

**Table 1. Pollen (XAC) and meteorological (XEMA) surface stations used in this study. Δkm is the distance between XAC and XEMA stations.**

| XAC site | Lat. (ºN), Lon. (ºE) | XEMA site | Lat. (ºN), Lon. (ºE) | Δkm |
|---|---|---|---|---|
| **Barcelona** | 41.39, 2.16 | Barcelona | 41.38, 2.16 | 5 |
| **Bellaterra** | 41.50, 2.10 | Sant Cugat | 41.48, 2.08 | 3 |
| **Girona** | 41.98, 2.82 | Girona | 41.98, 2.80 | 1 |
| **Lleida** | 41.62, 0.59 | Lleida | 41.59, 0.64 | 5 |
| **Manresa** | 41.72, 1.83 | Manresa | 41.70, 1.87 | 3 |
| **Roquetes** | 40.82, 0.49 | Aldover | 40.85, 0.50 | 4 |
| **Tarragona** | 41.12, 1.24 | Tarragona | 41.10, 1.20 | 4 |

**2.2. Airborne pollen columnar measurements**

The profiles of the particle backscatter coefficient and the volume and particle depolarization ratios were measured with the Barcelona Micro Pulse Lidar (MPL) system, Sigma Space model MPL-4B. The system is part of the MPLNET (Micro Pulse Lidar Network, http://mplnet.gsfc.nasa.gov/; Welton et al., 2018) network. The MPL system is a compact, eye-safe lidar designed for full-time unattended operation (Campbell et al., 2002; Flynn et al., 2007). It uses a pulsed solid-state laser emitting

low-energy pulses (~6 μJ) at a high pulse rate (2500 Hz), and a co-axial "transceiver" design with a telescope shared by both transmit and receive optics. The Barcelona MPL optical layout uses an actively controlled liquid crystal retarder which makes the system capable to conduct polarization-sensitive measurements by alternating between two retardation states (Flynn et al., 2007). The signals acquired in each of these states are recorded separately and called "co-polar" and "cross-polar". In nominal operation the raw temporal and vertical resolutions are 30 s and 15 m, respectively.

The total lidar signal, $P_T$, as a function of the altitude, $z$, is reconstructed from the "co-polar", $P_{co}$, and "cross-polar", $P_{cr}$, signals as (Flynn et al., 2007):

$$P_T(z) = P_{co}(z) + 2P_{cr}(z) \tag{1}$$

The particle backscatter coefficient, $\beta^p$, was retrieved with the two-component elastic algorithm (also known as the Klett-Fernald-Sasano method; Fernald, 1984; Sasano and Nakane, 1984; Klett, 1985) with a constant lidar ratio of 50 sr and applied

to the total lidar signal, $P_T$.

By adapting the notations of Flynn et al. (2007) to ours one can formulate the linear volume depolarization ratio, $\delta^V$, for the MPL system as:

$$\delta^V(z) = \frac{P_{cr}(z)}{P_{co}(z) + P_{cr}(z)} \tag{2}$$

The linear particle depolarization ratio, $\delta^p$, can then be determined by Eq. (4) of Sicard et al. (2016a). Finally, the extraction

of the total pollen backscatter coefficient, $\beta_{pol}$, out of the particle backscatter coefficient is made thanks to the pollen depolarization capabilities (see Sicard et al. (2016a) and references therein).



We also calculated the vertical height, $h_{pol}$, up to which the pollen plume extends. As it is shown in Sicard et al. (2016a), the pollen plume is characterized during the entire pollination event by a near-constant or slightly decreasing profile of $\beta_{pol}$. From this aspect the structure of the pollen plume is much simpler than the atmospheric boundary layer structure usually found in Barcelona (Sicard et al., 2006) and allows us to use a simple threshold method (Sicard et al., 2016a).

## 3. Modelling

The dispersion of the airborne pollen in the atmosphere was modelled with the Multiscale Online Nonhydrostatic AtmospheRe CHemistry model (MONARCH; Pérez et al., 2011; Jorba et al., 2012; Badia and Jorba, 2015; Badia et al., 2017). The MONARCH model is a fully online multiscale chemical weather prediction system for regional and global-scale applications with telescoping nest capabilities developed at the Barcelona Supercomputing Center (BSC). The system is based on the meteorological Nonhydrostatic Multiscale Model on the B-grid (NMMB; Janjic and Gall, 2012), widely verified at the National Center for Environmental Prediction (NCEP). The MONARCH model couples online the NMMB with the gas-phase and aerosol continuity equations to solve the atmospheric chemistry processes in detail. The model is designed to account for the feedbacks among gases, aerosol particles and meteorology. Currently, it can consider the direct radiative effect of aerosols while neglecting dynamic cloud–aerosol interactions. Different chemical processes were implemented following a modular operator splitting approach to solve the advection, diffusion, chemistry, dry and wet deposition, and emission of atmospheric constituents. Meteorological information is available at each time step to solve the chemistry. In order to maintain consistency with the meteorological solver, the chemical species are advected and mixed at the corresponding time step of the meteorological tracers using the same numerical schemes implemented in the NMMB. The advection scheme is Eulerian, positive definite and monotone, maintaining a consistent mass conservation of the chemical species within the domain of study (Janjic and Gall, 2012).

In this work, the model has been enhanced with a new pollen module that allows the study of the lifecycle of different pollen types. The numerical schemes used for the aerosols have been extended for pollen. The pollen type largely predominant during the pollination event analyzed in this study was *Pinus* (see Sections 1 and 5.1). Taking into account the study area considered in this paper (Figure 1a) and the pines in the surrounding territory that will be the main source providing pollen (Region V in CEFI, Gracia et al., 2000-2004), *Pinus* is present in 70.645 ha and the main species are *P. pinea* (accounting for 40,5% of the total surface of this species in Catalonia), *P. halepensis* (21,5%), and *P. sylvestris* (2,1%). The latter is not considered in the present work, as it pollinates later in the season not covered by the period under study. In this Section, we describe the pollen module and the setup of the numerical experiments.

### 3.1. Pollen representation: *Pinus*

The MONARCH model implements a mass-based aerosol scheme that has been extended to pollen bioaerosols. Table 2 summarizes the main characteristics of the implementation. *Pinus* pollen is represented as a spherical particle with a geometric diameter, $D_{Pinus}$, of 59 µm (Jackson and Lyford, 1999) and a dry mass density, $\rho_{Pinus-dry}$, of 560 kg m$^{-3}$ (Jackson and Lyford, 1999). The shape of *Pinus* grains is not completely spherical but for the sake of simplicity we make this assumption in the model. The aerosol life cycle is strongly affected by the water uptake. In our implementation, we consider *Pinus* pollen as an hydrophilic particle. Griffiths et al. (2012) describes the impact of relative humidity on the pollen density and radius. Following their results, we have implemented an increase of the *Pinus* pollen density with the relative humidity while the diameter of the particle is considered constant and independent of the water uptake. This effect is introduced in the model by considering relative humidity-prescribed growth factors ($\phi$) from Griffiths et al. (2012) shown in Table 3. Thus, the density of the particle is computed as:





$$\rho_{Pinus} = \rho_{Pinus-dry}\phi^{-3} + (1 - \phi^{-3})\rho_{Water} \tag{3}$$

where $\rho_{Pinus}$ is the mass density of the pollen with water uptake and $\rho_{Water}$ is the density of the water. $\rho_{Pinus}$ is used in the calculation of sedimentation, dry deposition, and wet deposition. The numerical schemes for all these processes are described in Pérez et al. (2011).

5    **Table 2. Databases and *Pinus* pollen parameters used in the pollen scheme.**

|  | Value or Range | Source |
|---|---|---|
| **Geographical distribution** | - | Cartography of habitats of Catalonia - Carreras et al. (2015) |
| **Tree density (tree ha⁻¹)** | 484 - 957 | Forest Inventory of Catalonia - Gracia et al. (2000-2004) |
| **Emission factor ($10^{10}$ grain tree⁻¹ season⁻¹)** | 25.1 | Tormo et al. (1996) |
| **Grain diameter (µm)** | 59 | Jackson and Liford (1999) |
| **Grain mass density (kg m⁻³)** | 560 | Jackson and Liford (1999) |

**Table 3. Growth factor ($\phi$) of *Pinus* pollen for ranges of ambient relative humidity.**

|  | **Growth factor $\phi$** |
|---|---|
| **90 % < Relative humidity** | 1.7 |
| **80 % < Relative humidity < 90 %** | 1.28 |
| **70 % < Relative humidity < 80 %** | 1.16 |
| **50 % < Relative humidity < 70 %** | 1.07 |
| **Relative humidity < 50 %** | 1.0 |

### 3.2.  Emission scheme

The emission scheme implemented in MONARCH is based on the concepts of the parameterization of Helbig et al. (2004)
10   with some modifications. It computes the vertical emission flux of *Pinus* pollen grains per grid cell $x$ and unit time $t$, as:

$$E_{Pinus}(x,t) = P(x) \cdot F(t) \cdot R(x,t) \cdot C \tag{4}$$

where $E_{Pinus}$ is the vertical emission flux of *Pinus* pollen (kg·m⁻²·s⁻¹), $P$ is a characteristic mass concentration of the pollen grains available in the canopy of the pine trees (kg·m⁻³), $F$ is the unitless phenology function scaling $P$ through the pollination event with values ranging from 0 to 1, $R$ is the weather-dependent release scaling function (m·s⁻¹) which depends on
15   instantaneous meteorological conditions, and $C$ is a global calibration factor accounting for the uncertainty of the model schemes. Note that $P$ is not the actual pollen released in the atmosphere by the tree; this will depend on the environmental conditions that may favor the release of the pollen grains from the tree described by $R$.



The available number of pollen grains per tree during a season is a highly uncertain parameter. Several works provide estimates of this parameter for pines. Tormo et al. (1996) calculated the annual production of pollen grains for three individual pine trees (*P. pinaster*) in good health and well-shape located in southwestern Spain obtaining values of 20.9, 32.3 and 22.2 billions of pollen grains. Parker and Blush (1996) reported, for an open-grown *P. taeda* tree, a production of 8 billions of pollen grains

during a season. And Williams (2008) emphasized the fact that pollen production depends on the tree age and the size of the crown, increasing with both. In this work, we use the average of Tormo et al. (1996) for *P. pinaster* as representative estimate of the emission of type of pine trees present in the area of interest though being a highly uncertain parameter. Thus, we formulate $P$ as:

$$P(x) = \frac{Grain \cdot M}{\gamma \cdot H_S} \qquad (5)$$

where $Grain$ is the production of pollen grains in mass per tree during the season (kg·tree$^{-1}$), $M$ is the number of trees in a grid cell, $H_S$ is the average canopy height for *Pinus* trees (m), and $\gamma$ is the area covered by the trees in a grid cell (m$^2$). $Grain$ is the average of Tormo et al. (1996) results, 25.1 billions of pollen grains, transformed in mass using the *Pinus* grain size and density assumed in Section 3.1 Different values of $H_S$ are found in the literature for pine trees. The Forest Inventory of Catalonia (Gracia et al., 2000-2004) provides an average height of 11.3 m for the area of study, which is the value used in our work. No

detailed information is available to describe $\frac{M}{\gamma}$ for the domain of study, we approximate the term with available information of the *Pinus* tree density as follows. The spatial distribution of the pine species of interest in our study that pollinate from February to April in Catalonia (*P. halepensis*, *P. pinea)* is obtained from the Cartography of habitats of Catalonia (Carreras et al., 2015). The cartography has been remapped to 1 km resolution dataset and then combined with the pine tree density reported by the Forest Inventory of Catalonia (Gracia et al. 2000-2004). The pine density in Catalonia ranges between 484 to 957 trees·ha$^{-1}$.

With this information, a database with the number of pine trees per grid cell of the model is derived to estimate $M$ (see Figure 1a). This dataset is complemented with the inventory of pine trees of the Barcelona's City Council that provides specific georeferenced information of the trees of the city.

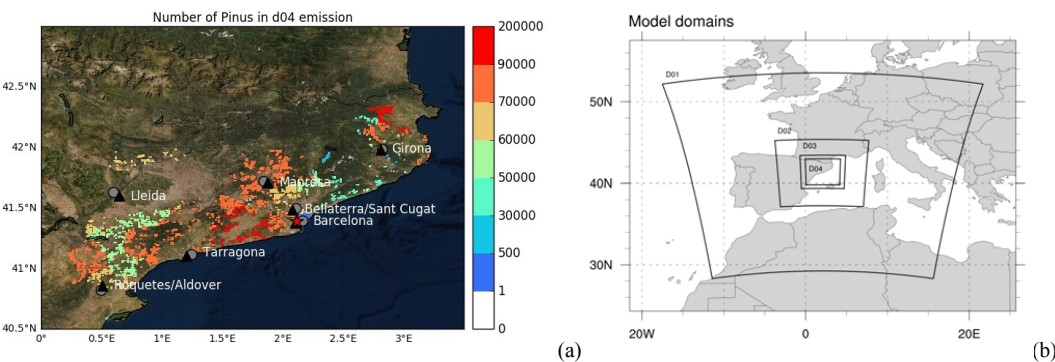

**Figure 1. (a) Pine tree density (tree km$^{-2}$) in the Catalonia region at 1km resolution. The stations of the Aerobiological Network of**
**Catalonia (XAC) used in this work are depicted in grey circles, the meteorological stations of the Automatic Weather Stations Network of Catalonia (XEMA) in black triangles, and the Lidar station in red star. Names of the sites next to the symbols in the order XAC/XEMA in cases where stations are very close. (b) Simulation domains: D01 at 27 km, D02 at 9km, D03 at 3km, and D04 at 1km horizontal resolution. Basemap source: Esri (2020).**

Once $P$ is estimated, the emission scheme needs a function that describes how the availability of pollen grains in the tree
evolves during the pollination season. This function is known as the phenology function, $F$. Not all the pollen that the tree will release during a season is available the first day of the pollination; this will evolve during the season depending again on the meteorological and climatological conditions that have affected the plant during the last year. Several efforts have been done to develop phenology functions for different types of plants based on relationships between pollen counts and key





meteorological factors (Jones and Harrison 2004; Schueler and Schlünzen, 2006; Laursen et al., 2007; Linkosalo et al., 2010; Marceau et al., 2011), and have been used in regional models to predict airborne concentrations of different types of pollen (Helbig et al., 2004; Sofiev et al., 2006; 2013; Zink et al., 2013; Zhang et al., 2014; Sofiev et al., 2017). All current schemes show limitations in the prediction of the onset and duration of the pollen season and may have significant biases for very

localized and specific pollen events as the one we are studying in this work. In this sense, we decided to implement a much simpler function that is mainly constrained by the observed pollen counts in the aerobiological station of Barcelona. The evolution of the pollen concentrations during the period 1 March – 30 April, 2015, have been fitted with a Gaussian function that describes the evolution of the event (Figure 2). A similar approach has been used in other works (e.g., Wozniak and Steiner, 2017). The resulting normalized function has been selected as the phenology function $F$ of the emission scheme. With

this simple approach, we assure that the emission scheme will release an amount of pollen in the atmosphere reasonable to reproduce the observed pollen concentrations in the atmosphere retrieved by the lidar and the Hirst collectors in the region of Barcelona.

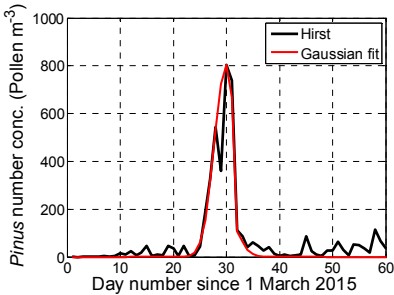

**Figure 2.** *Pinus* pollen daily concentration measured with a Hirst collector at the Barcelona site (black line) during March and April
2015 and fitted with a Gaussian curve (red line) used as phenology function, $F(t)$, in the emission scheme of the model.

The weather-dependent function $R$ implemented in the model accounts for the mobilization effect of the wind and turbulence of the near surface layer of the atmosphere (Helbig et al., 2004):

$$R(x,t) = K_e \cdot u_* \tag{6}$$

where $K_e$ is a wind effect scale factor between 0 and 1 (unitless) and $u_*$ is the friction velocity (m·s$^{-1}$). $K_e$ is parameterized
using a threshold friction velocity, $u_{*te}$, as:

$$K_e = \begin{cases} 1 - \dfrac{u_{*te}}{u_*}, u_* > u_{*te} \\ 0, u_* \le u_{*te} \end{cases} \tag{7}$$

$$u_{*te} = \alpha \cdot u_{*t} \tag{8}$$

The threshold friction velocity is the product of a resistance term, $\alpha$, and a standard threshold friction velocity, $u_{*t}$ (m·s$^{-1}$). The expression of $u_{*t}$ is the regression formula of Greeley and Iversen (1985) based on wind tunnel data for sand erosion. The

resistance term $\alpha$ is used to distinguish the different natures of sand erosion on the ground and the pollen release above the canopy height, and it is defined as:

$$\alpha = \dfrac{u_{10e}}{u_{10}} \tag{9}$$

where $u_{10e}$ is an empirical threshold wind speed for type of tree (m·s$^{-1}$), and $u_{10}$ is the 10-m wind speed of the model (m·s$^{-1}$). No specific values of $u_{10e}$ are found in the literature for pine trees; here we set $u_{10e}$ to 2.9 m·s$^{-1}$ using the value of Helbig et
al. (2004) for Alder. Zhang et al. (2014) did not find significant sensitivity to different values of $u_{10e}$.



### 3.3. Model setup and runs

To study the dispersion of *Pinus* pollen in the region of study, the MONARCH model has been configured with 4 domains using the telescoping nest capability of the system (see Figure 1b). A parent domain covering central Europe and centered over the region of interest is set with 27 km horizontal resolution and 48 vertical layers with the top of the atmosphere at 50 hPa. Three nested domains are defined to increase the model resolution up to 1 km in the innermost domain with a ratio of 1:3 from one nest to the other.

The simulations cover the period 20 March to 2 April 2015, which comprises the pollination event under study in this work. The meteorological initial and boundary conditions are obtained from the ERA-5 reanalysis at 30 km horizontal resolution and 6 hour frequency. No boundary conditions for the *Pinus* pollen are prescribed in the parent domain as the contribution from long-range transport is considered minor compared with the local emissions. The model outputs the *Pinus* pollen mass concentration over the domain of study with hourly resolution which is converted to number concentration as a diagnostic.

Note that the *Pinus* pollen mask used in all domains is the one described in Section 3.2 and only considers the trees of the Catalonia region pollinating during March and April. The main objective of the work is to assess the vertical distribution of pollen nearby the Barcelona site where the lidar is available. We consider that the *Pinus* pollen detected by the lidar and collected by the Hirst instruments are mainly from local origin, as the size of the pollen is significantly coarse and limits the long-range transport of this type of bioaerosol. To explore this initial hypothesis two different runs were conducted: (1) a *Basecase* simulation using the model parameters described in Section 3.1, and (2) an *Enhanced* simulation where the lifetime of the *Pinus* pollen is increased by assuming that the sedimentation and dry deposition of the pollen is half of the estimated in the *Basecase*. A global calibration factor $C$ of 2.9 was used to minimize the error of the *Basecase* simulation with respect to the Hirst observations. Other works use $C$ as a 2D spatial factor to calibrate the emission flux better due to the large uncertainties of most pollen emissions schemes and input datasets (e.g., Kurganskiy et al., 2020). In Section 5, the model results for both scenarios are discussed with the Hirst and lidar measurements.

### 4. Conversion of lidar- and model-derived magnitudes to number concentration

We propose to perform a closure study between in-situ and column measurements by constraining the lidar retrieval in the first part of the vertical profile to the concentration measurement at ground level. An important added value is that the closure study also fixes parameters needed for the conversion of the model output. In order to compare the lidar retrieval to measurements made at ground level, we consider the first lidar measurement (225 m) to be a proxy of what it would be at ground level. This hypothesis is somehow validated by the fact that the lidar vertical distribution, as it will be shown later, is rather flat (concentrations barely vary) as one gets closer to the ground. In the case of the model, we consider the first model layer, the center of which varies between 24 and 24.4 m over all the simulations considered here.

### 4.1. Definition of magnitudes and conversion formulas

From the analysis of the samples of the Hirst collectors the concentration of the pollen type $i$, $C_{,i}^H$ (Pollen m$^{-3}$), is obtained. The manual procedure to do so include neither annotating information on the size of the pollen grain, nor any kind of information on its optical properties. This observation is taken as the reference value to which the lidar retrieval is constrained and to which the model output is compared.





The lidar instrument measures the particle backscatter coefficient from which the total pollen contribution, i.e. the total pollen backscatter coefficient, $\beta_{pol}$ (m⁻¹sr⁻¹), can be extracted as in Sicard et al. (2016a). The total pollen mass concentration, $C_m^{lid}$ (kg m⁻³), is obtained using the following relationship:

$$C_m^{lid} = \frac{\beta_{pol}LR}{\sigma^*} \tag{10}$$

where $LR$ (sr) is the lidar ratio and $\sigma^*$ (m²g⁻¹) the specific extinction cross-section. The latter is actually not a fixed constant, as it will depend on the taxa present, their concentration and weight, as well as their optical properties. It is thus virtually unknown.

The mass concentration of Eq. (10) can be converted into number concentration of the specie $i$, $C_{,i}^{lid}$ (Pollen m⁻³), as:

$$C_{,i}^{lid} = \frac{C_m^{lid}wf(i)}{\frac{4\pi}{3}\left(\frac{D(i)}{2}\right)^3\rho(i)} = \frac{\beta_{pol}LRwf(i)}{\sigma^*\frac{4\pi}{3}\left(\frac{D(i)}{2}\right)^3\rho(i)} \tag{11}$$

where for each specie $i$ $wf(i)$ is the weight factor, $D(i)$ (m) the diameter, and $\rho(i)$ (kg m⁻³) the mass density. It is important to note that in this last conversion the volume of a pollen grain is calculated assuming that the grain has a spherical shape in agreement with the assumption made in the model. $wf(i)$ represents the fraction of the mass of the specie $i$ to the total pollen mass. It is calculated on a daily basis using the daily concentration measurements of the Hirst collectors as:

$$wf(i) = \frac{C_{,i}^H\frac{4\pi}{3}\left(\frac{D(i)}{2}\right)^3\rho(i)}{\Sigma_u\left[C_{,u}^H\frac{4\pi}{3}\left(\frac{D(u)}{2}\right)^3\rho(u)\right]} = \frac{C_{,i}^HD^3(i)\rho(i)}{\Sigma_u[C_{,u}^HD^3(u)\rho(u)]} \tag{12}$$

In Eq. (11) the product $C_m^{lid}wf(i)$ represents the mass concentration estimated from the lidar measured total backscatter coefficient for the specie $i$ and is directly comparable to the model output. This implies that the choice of $D(i)$ and $\rho(i)$ has no impact on the comparisons of the lidar and the model.

Finally, the model provides mass concentration for the specie $i$, $C_{m,i}^{mod}$ (kg m⁻³). The number concentration of the specie $i$ is then calculated as:

$$C_{,i}^{mod} = \frac{C_{m,i}^{mod}}{\frac{4\pi}{3}\left(\frac{D(i)}{2}\right)^3\rho(i)} \tag{13}$$

The next step is to optimize all parameters to minimize the difference between the two observations $C_{,i}^H$ and $C_{,i}^{lid}$, and then apply these optimum parameters to compute $C_{,i}^{mod}$ and compare it to both $C_{,i}^H$ and $C_{,i}^{lid}$.

### 4.2. Numerical values used in this work

The lidar number concentration, $C_{,i}^{lid}$, for the specie $i$ depends thus on 4 parameters: $LR$, $\sigma^*$, $D(i)$ and $\rho(i)$, while the model
number concentration for the specie $i$ depends only on 2: $D(i)$ and $\rho(i)$. The conversion of the lidar retrieval into number concentration has 3 degrees of freedom, while the conversion of the model output has only 1.

Each day, for the 24 hours ($t$ = 1, 2, 3, …, 24) between 00 and 23UTC, we use the least-squares method to obtain the optimal specific extinction cross-section by minimizing the sum of squared residuals expressed by:

$$\Sigma_t[C_{,i}^H(t) - C_{,i}^{lid}(t)]^2 = \Sigma_t\left[C_{,i}^H(t) - \frac{\beta_{pol}(t)LRwf(i)}{\sigma^*\frac{4\pi}{3}\left(\frac{D(i)}{2}\right)^3\rho(i)}\right]^2 \tag{14}$$

This minimization can be done in terms of $LR$, $\sigma^*$, $D(i)$ or $\rho(i)$. As the greatest unknown is without any doubt $\sigma^*$, all other parameters ($LR$, $D(i)$ and $\rho(i)$) are fixed. However we also perform a sensitivity analysis on these parameters to quantify the





range of uncertainty associated to $\sigma^*$ (see Section 4.3). The minimization of Eq. (14) yields the best estimate solution in terms of $\sigma^*$:

$$\sigma^* = \frac{LRwf(i)}{\frac{4\pi}{3}\left(\frac{D(i)}{2}\right)^3 \rho(i)} \frac{\Sigma_t[\beta_{pol}(t)]^2}{\Sigma_t\left[C_{,i}^H(t)\beta_{pol}(t)\right]} \qquad (15)$$

The weight factor $wf(i)$ for *Pinus* pollen is calculated as the daily mean of the hourly fractions of mass concentration presented

in Sicard et al. (2019). Daily values of $wf$ are reported in Table 4. The mean value over the whole event is 0.901, which is equal to say that *Pinus* pollen represents 90.1 % of the mass of total pollen. $LR$ is set to 50 sr, a reasonable value for pollen justified in Sicard et al. (2016a). *Pinus* pollen diameter $D_{Pinus}$ and mass density $\rho_{Pinus}$ are taken from Jackson and Lyford (1999) and are set to 59 µm and 560 kg·m⁻³, respectively, in agreement with the model (see Section 3.1). With these values set, we calculate $\sigma^*$ with Eq. (15) and find the daily values reported in Table 4. Values of $\sigma^*$ oscillate between 0.78 and 1.67

m²g⁻¹. A factor of almost 2 between the minimum and maximum values is not that surprising taking into account the lengthy methodology followed, and in particular the extraction of the contribution of pollen out of the total backscatter coefficient and the hypothesis made that the first lidar measurement (at 225 m) is a good proxy of what it would be at ground level. As an example, if we compare our pollen values of $\sigma^*$ with the ones of other large particles, we find that our values are larger than mineral dust observed in southern Europe which has values on the order of 0.5 – 0.6 m²g⁻¹ (Pérez et al., 2006), and they are

smaller than values for sea salt surrogate estimated to be on the order of 6 m²g⁻¹ (Radney et al., 2013). These results are consistent since pollen grains are much less dense in mass (and thus have a larger $\sigma^*$) than mineral dust, and because of the strong scattering properties (and thus large $\sigma^*$) of sea salt.

As an illustration of how lidar and Hirst number concentration agree with the values of $\sigma^*$ found, a scatterplot of the lidar vs. Hirst concentration on 27 March ($\sigma^*$ =1.21 m²g⁻¹), as well as the daily cycle of both concentrations are shown in Figure 3. 27

March is chosen because, according to Sicard et al. (2016a), it is the day when all column properties correlate the best with the *Pinus* pollen surface concentration. The scatterplot is rather disperse around the 1:1 line (Figure 3a), hence the correlation coefficient, equal to 0.55, is not very high. One sees that both lidar and Hirst concentration daily cycles agree relatively well (Figure 3b) except in the very first hours of the day when no pollen is detected in the column whereas it is present at the surface.

**4.3. Sensitivity study on the specific extinction cross-section $\sigma^*$**

In order to quantify for the pollen type of interest in this study, *Pinus*, the sensitivity of $\sigma^*$ to the rest of parameters, $LR$, $D_{Pinus}$ and $\rho_{Pinus}$, we vary these latter and resolve Eq. (15). $LR$, $D_{Pinus}$ and $\rho_{Pinus}$ are varied around their nominal value (see former Section) as follows: 50 ± 10 sr, 59 ± 10 µm and 560 ± 200 kg·m⁻³, respectively. Although the estimation of $\sigma^*$ is made on a daily basis, it is not necessary to perform the sensitivity study for the five days of the event since the relative variations of $\sigma^*$

as a function of the same $\Delta LR$, $\Delta D_{Pinus}$ and $\Delta\rho_{Pinus}$ from one day to another will not be different. The sensitivity analysis is thus performed for 27 March, the day when all column properties correlate the best with the *Pinus* surface concentration (Sicard et al., 2016a). Values of $\sigma^*$ are reported in Table 5 and its dependency to the rest of parameters is shown in Figure 4. Obvious results are that $\sigma^*$ decreases with decreasing $LR$ and increasing $D_{Pinus}$ and $\rho_{Pinus}$. The strongest sensitivity of $\sigma^*$ is observed for $D_{Pinus}$ (for $LR$ and $\rho_{Pinus}$ equals to their nominal values, $\sigma^*$ varies between 0.76 and 2.12 m²g⁻¹ when $D_{Pinus}$ =

59±10 µm), while the lowest one is observed for $LR$ (for $D_{Pinus}$ and $\rho_{Pinus}$ equals to their nominal values, $\sigma^*$ varies between 0.97 and 1.45 m²g⁻¹ when $LR$ = 50±10 sr). In these conditions, when all parameters are fixed to their nominal values, $\sigma^*$ will vary by -0.45 / +0.91 m²g⁻¹ (-37 / +75 %) around its nominal value of 1.21 m²g⁻¹ if $D_{Pinus}$ differs from its nominal value by ± 10 µm; it will vary by -0.32 / +0.68 m²g⁻¹ (-26 / +56 %) around its nominal value if $\rho_{Pinus}$ differs from its nominal value by ± 200 kg·m⁻³; and it will vary ± 0.24 m²g⁻¹ (± 20 %) around its nominal value if $LR$ differs from its nominal value by ± 10 sr. In





the worst case, when all parameters are relaxed, $\sigma^*$ could vary down to 0.45 (-63 % w.r.t. its nominal value) and up to 3.95 (+226 % w.r.t. its nominal value). This worst scenario is however very unlikely to happen for various reasons. First, all parameters ($LR$, $D_{Pinus}$ and $\rho_{Pinus}$) should take their extreme values simultaneously and this is rather improbable. Second, the probability of $LR$ taking values of 40 (50 – 10) or 60 (50 + 10) sr is very low, as suggested by Noh et al. (2013) who found a

mean columnar lidar ratio of 50 sr and a standard deviation not greater than 6 sr during a 6-day pollination event (mostly dominated by *Pinus* and *Quercus* pollen) in South Korea. Third, in terms of grain diameter, although there is a large range of *Pinus* grain diameter measured in different regions of the planet (46 μm, Japanese black pine, Japan (Hirose and Osada, 2016); 51 ± 7 μm, *P. sylvestris*, China (Song et al., 2012); 52 μm, *P. sylvestris*, US (Durham, 1946); 55 ± 4 μm, *P. sylvestris*, Finland (Varis et al., 2011); 51 – 100 μm, *P. sylvestris*, Austria (Halbritter, 2016)), the species considered in this study (*P. halepensis*

and *P. pinea*) have a narrower range of mean diameters. Roure (1985) measured diameters of these species in the Iberian Peninsula of 41–55, 43–65 and 47–65, respectively. The range explored in our sensitivity study, 49 – 69 μm, corresponds to ± 10 μm around the value of 59 μm measured for *P. sylvestris* by Dyakowska and Zurzycki (1959) and referenced in Jackson and Lyford (1999). Fourth and last, *Pinus* grain density measurements are pretty seldom in the literature. Our nominal value of 560 kg·m⁻³ was taken from Jackson and Lyford (1999). Other values of 450 kg·m⁻³ have been estimated for *P. sylvestris* in

the US (Durham, 1946) and of 550 kg·m⁻³ for Japanese black Pine in Japan (Hirose and Osada, 2016). The range explored in our sensitivity analysis, 360 – 760 kg·m⁻³, is thus conservative and extreme values of 360 and 760 kg·m⁻³ are rather very unlikely to be reached.

**Table 4. Summary of the daily values of *wf* and $\sigma^*$.**

|  | 27 March | 28 March | 29 March | 30 March | 31 March |
|---|---|---|---|---|---|
| ***Pinus* pollen *wf*** | 0.895 | 0.875 | 0.915 | 0.915 | 0.904 |
| **$\sigma^*$ (m²g⁻¹)** | 1.21 | 0.99 | 1.67 | 1.44 | 0.78 |

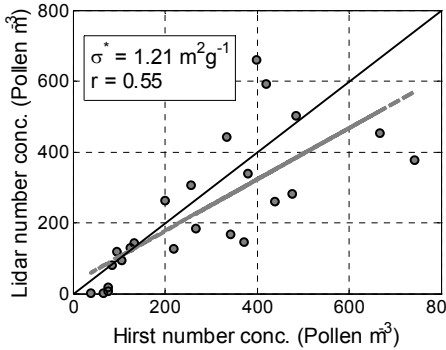

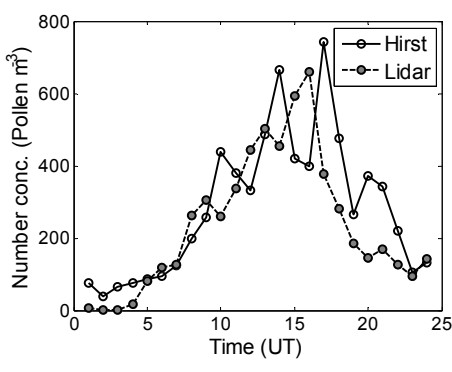

(a)                                    (b)

**Figure 3. (a) Scatterplot of lidar vs. Hirst *Pinus* number concentration at or close to the ground level on 27 March with $\sigma^*$ =1.21 m²g⁻¹; (b) daily cycle of both concentrations. In plot (a) *r*=0.55 is the correlation coefficient.**

## 5.     Results

### 5.1.   General overview and horizontal representativeness of the model

The event of interest took place between 27 and 31 March 2015, the most intense period of the *Pinus* pollination season in the

region (Sicard et al., 2016a). The five days of the event are noted in the following 27M, 28M, 29M, 30M and 31M. A detailed





analysis of the event is presented in Sicard et al. (2016a). In the second half of March 2015 a strong anticyclone positioned in the Atlantic Ocean west of the Portuguese coast generated northwesterly winds in the northeastern part of the Iberian Peninsula. The event was characterized by medium-strong winds in the medium troposphere (500hPa) veering from north to northwest during the 5 days of the period and resulting with northwesterly winds near-surface and clear skies in Barcelona most of the

time. Figure 5a shows the hourly number concentration of the total pollen and *Pinus* pollen measured in Barcelona downtown with the Hirst during the event. Total pollen hourly concentrations reached values higher than 6000 Pollen m$^{-3}$ on 31M. Every day the *Pinus* pollen shows a diurnal cycle with several concentration peaks of major intensity. Figure 5b represents the fraction of the mass concentration of *Pinus* pollen calculated according to Sicard et al. (2019). *Pinus* pollen represents 90 % of the mass of total pollen. As *Pinus* pollen grains are much larger than the rest of the pollen types present during this event, their

contribution in mass dominates. In this work, we focus the analysis in the *Pinus* pollen because it is the taxon with the major contribution to the total mass of pollen measured in Barcelona, and the emission source is widely distributed across the domain of interest (see Figure 1a). The latter allows a better analysis of the role of short-to-medium range transport and vertical mixing processes in the dispersion of this type of pollen.

**Table 5.** Sensitivity study on $\sigma^*$ (m$^2$g$^{-1}$) produced by controlled variations of *LR*, $D_{Pinus}$ and $\rho_{Pinus}$. **The nominal set of values used**
**in the rest of this article is indicated in bold font.**

|  |  | LR = 40 sr | | | LR = 50 sr | | | LR = 60 sr | | |
|---|---|---|---|---|---|---|---|---|---|---|
|  |  | $\rho_{Pinus}(kg \cdot m^{-3})$ | | | | | | | | |
|  |  | 360 | 560 | 760 | 360 | 560 | 760 | 360 | 560 | 760 |
| $D_{Pinus}$ ($\mu m$) | 49 | 2.63 | 1.69 | 1.25 | 3.30 | 2.12 | 1.56 | 3.95 | 2.54 | 1.87 |
| | 59 | 1.51 | 0.97 | 0.71 | 1.89 | **1.21** | 0.89 | 2.26 | 1.45 | 1.07 |
| | 69 | 0.94 | 0.61 | 0.45 | 1.18 | 0.76 | 0.56 | 1.41 | 0.91 | 0.67 |

Figure 5c also shows the quicklook, also called time-height plot, of the volume depolarization ratio of the lidar. It is not a quantitative plot since both the molecules and the particles contribute to the volume depolarization but it is an indicator of the presence or not of depolarizing particles (here, pollen grains). In Figure 5c the dark green areas represent molecular level (only detectable at the high temporal resolution of the quicklook during nighttime because of the reduced background signal as

opposed to daytime), light green areas represent low-depolarizing particles (urban background) and yellow/orange areas represent high-depolarizing particles, i.e. pollen. The vertical distribution of the airborne pollen also shows a clear diurnal cycle with usually no or weak nighttime activity in the upper layers, the pollen grains probably staying very close to the ground. The diurnal cycle is marked by an increase in amplitude and height of the volume depolarization ratio starting around 10UTC and a decrease staring before 16UTC. This diurnal pattern is observed on each single day of the pollination event. On the first

4 days the volume depolarization ratio has come back to its background value (< 0.08) before 20UTC. Most of the aerosol load is usually found below 1.5 km. While the peak of *Pinus* pollen hourly concentration at the surface (1266 Pollen m$^{-3}$) is reached on 30M at 08 – 09UTC, the peak in the column (quantified by the pollen aerosol optical depth (AOD) in Sicard et al. (2016a)) is reached 4 hours later at 12 – 13UTC. Note that even though weak or no nigthttime activity is detected by the lidar, the ground measurements indicate that periods with significant nighttime pollen concentrations (e.g. the first hours of 28M

and 31M) are present. This is a good indication of the stratification of the pollen in the lowermost layers during nighttime.

In order to understand the dynamics of *Pinus* pollen transport during the event, we present here the model results over the innermost domain (D04) at 1 km horizontal resolution of the *Basecase* configuration described in Section 3 The *Pinus* number



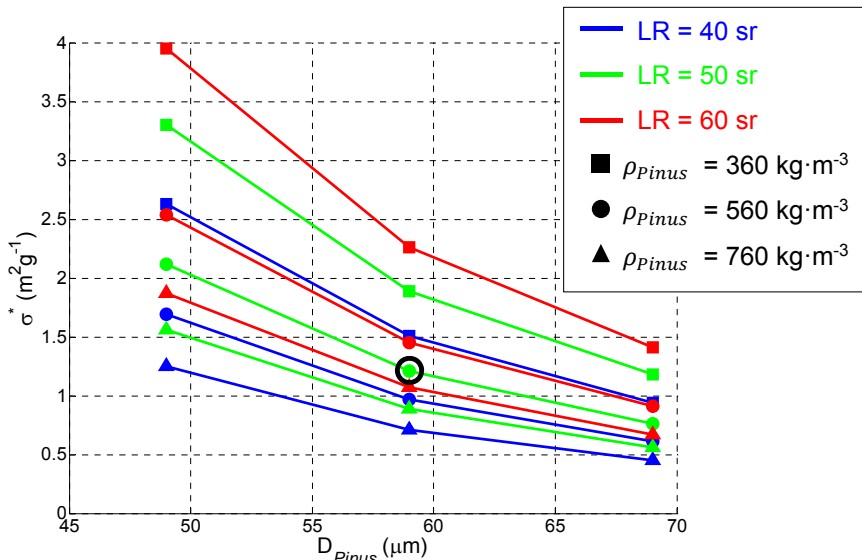

**Figure 4. (a) Sensitivity study of $\sigma^*$ to *LR*, $D_{Pinus}$ and $\rho_{Pinus}$ performed on 27 March. The nominal set of values used in the rest of this article is indicated with a black circle.**

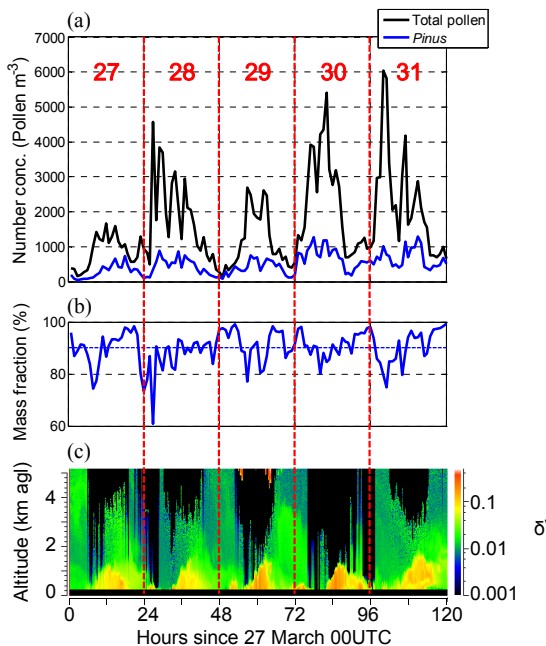

**Figure 5. (a) Total pollen and *Pinus* number concentration; (b) Fraction of mass concentration of *Pinus* pollen; (c) Vertical distribution of the lidar volume depolarization ratio, $\delta^V$, as a function of time. The period goes from 27M at 00:00UTC until 31M at 23:59UTC.**

concentration at surface (first model layer, around 24 m a.g.l.) from 27M to 31M is shown in Figure 6. The entire event is characterized by a dispersion of pollen grains towards the Mediterranean Sea, mainly driven by the northwesterly winds. The highest concentrations are in the southwest region of the domain. This area is dominated by the natural channelization of the Ebro Valley, and a characteristic northwest wind, called Cierzo, develops under west-northwest advection situations. Regular





outbreaks of plumes with high concentrations of pollen are transported hundreds of kilometres to the Mediterranean Sea. This effect is less pronounced towards the north of the domain. Note that the regions with pines are found within the first 150 km inland from the coast (see Figure 1a); beyond that distance *Pinus* trees are scarce. Three main areas are identified from the pine tree density map used in the model: (1) in the south, a wide area with medium-high pine density are present, (2) in the

central coast, the area with pines is even more extensive with high densities of trees, and (3) in the northern coast, the presence of pines is more reduced but still some localized regions with high densities of trees are present. This tree distribution and the strong winds in the southwest of the domain explain the spatial gradient in concentrations simulated by the model.

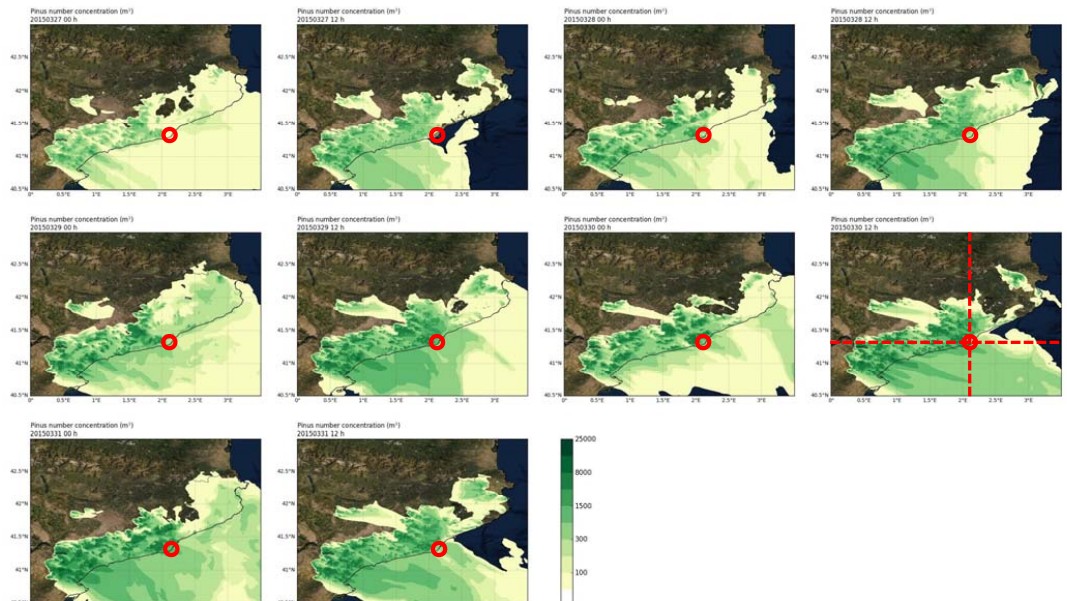

**Figure 6. Maps of the surface *Pinus* number concentration (Pollen m$^{-3}$) simulated at 1 km resolution (D04) at 00UTC and 12UTC**
**for the period 27M to 31M in green colour scale. The red circles indicate the location of Barcelona. Red dashed lines indicate the vertical cross sections shown in Figure 10. Basemap source: Esri (2020).**

To assess the representativeness of the model results, we have compared them with the hourly meteorological and daily aerobiological observations described in Section 2. Here, we focus on surface observations. The quality of the results is quantified by means of classical statistics (Pearson correlation coefficient, *r*, root mean square error, *rmse*, and bias, *bias*)
using 7 aerobiological stations of the XAC network and 7 meteorological stations of the XEMA network (details on the localization in Figure 1a and Table 1). The variables evaluated are hourly 2-meter air temperature, 10-meter wind speed and direction, irradiance, 2-meter relative humidity, and daily *Pinus* number concentration at the first model layer.

Regarding the meteorological results, Table 6 presents the statistics for the 9 km *Basecase* run and all 7 stations (results for 3 and 1 km in the Appendix A) and Figure 7 shows the temporal evolution of the same variables (except concentration) at the
three model resolutions for Barcelona and Bellaterra/Sant Cugat (results at the other 5 stations in Appendix B). The statistics indicates a good agreement for temperature, irradiance and relative humidity in most of the stations (correlations above 0.9 and low bias), while higher errors are observed in the winds. Overall, the results are within the typical performance range of mesoscale models in the area of study (i.e., Jiménez-Guerrero et al., 2008), being surface winds one of the most difficult variables to reproduce in coastal regions with complex terrain like the one under study. On the one hand, the model results in
coastal sites may show higher errors in temperature, winds, and relative humidity. These sites are close to the interface sea-land and small inaccuracies in the representation of the coastline may have a strong impact in the results. Tarragona site is a clear example, where the temperature results are degraded with the 1 km domain compared with the upper nests (see Appendix





A and B) as a result of the land and sea grid cells represented in each model resolution surrounding the site. An excessive influence of marine air masses will result in a lower thermal amplitude and overestimated relative humidity. On the other hand, better statistics are obtained at inland sites where the topography is properly captured by the model resolution. Most sites show *rmse* and *bias* below 2.6ºC and $\pm$1.2ºC, 2 m s$^{-1}$ and $\pm$0.5 m s$^{-1}$, 90 degrees and $\pm$11 degrees, and 15% and $\pm$6%, for temperature, wind speed, wind direction and relative humidity, respectively.

Barcelona site is representative of a coastal station, located few kilometres of the shore, while Bellaterra/Sant Cugat is an example of an inland site not affected by major mountain ranges (i.e., the Pyrenees). In both stations, an improvement is detected in most meteorological variables with the increase of resolution (see Figure 7). The model reproduces the daily cycle of the temperature and relative humidity, and captures the cloudiness observed during 29M and 30M. As noted in the statistics, more disagreement is seen in the wind speed, and the different resolutions show larger variability compared with the other variables. The wind results in Barcelona site are consistent among the model resolutions. Some systematic bias is observed in wind speed at the end of the day and during the first hours of the day after. The model overestimates the calm winds observed during nighttime and it tends to underestimate the morning peak (i.e., transitions from 27M to 28M, 29M to 30M and 30M to 31M). This results in lower relative humidity and higher temperatures during the calm periods compared with observations. Albeit this does not impact the wind direction that is well captured most of the time. The results at the Bellaterra/Sant Cugat site show a clear improvement with the resolution. The site is located in a long valley surrounded by two mountain ranges that are better represented in the model by increasing the resolution. The overestimated winds during nighttime in Bellaterra/Sant Cugat results in model winds veering from southwest to northwest (i.e., 27M and 29M), while observations show a calm-stagnated situation. Such error leads to a model underestimation of relative humidity and overestimation of temperature there. On the night of 31M, the observations show weak easterly winds while the model develops northwest winds. Albeit the limitations identified, the model captures the meteorology of the event under study reasonably well.

The main driver of the pollen release in the atmosphere is the wind. The period of study is not characterized by intense surface winds: wind speed peaks are below 10 m s$^{-1}$ (see Figure 7 and Appendix B). During periods of moderate winds (around 5 m s$^{-1}$) the release of pollen is significantly stronger, concentrations of 300-1500 Pollen m$^{-3}$ are simulated 200 km downwind the emission sources (see Figure 6) and concentrations over the Mediterranean Sea above 5000 Pollen m$^{-3}$ are not unusual (e.g., 29M at 12UTC). Under weak or calm conditions (wind speed below 1 m s$^{-1}$), very small concentrations are simulated downwind the sources (< 100 Pollen m$^{-3}$), while localized high concentrations are still present close to the regions with higher pine density. The strength of the winds decreases during nighttime, which reduces the dispersion of pollen and could stop the emission process. This might be explained by the presence of weak but non-negligible winds that promote some emission, an important reduction of the dispersion, and an accumulation of grains in the lower boundary layer due to a stable atmosphere.

The daily average of the *Pinus* surface concentration is evaluated in the XAC sites. Table 6 presents the statistics for the period 25 March to 2 April 2015 (all stations; 9 km domain), and Figure 8 the temporal evolution for the same period of Barcelona, Tarragona and Bellaterra sites (all three domains). The model performs significantly different at those three sites compared with the rest of locations. The three sites are characterized by an important upwind tree density (see Figure 1a and 6). On the one hand, the correlation is 0.5 in Barcelona, 0.6 in Tarragona, and 0.9 in Bellaterra, with a negative bias more pronounced in Bellaterra (-366 Pollen m$^{-3}$). In this site, measurements show very high concentrations, particularly on 30M, that the model does not capture in intensity, although it reproduces the evolution of the event. In fact the evolution of the event is captured in the three sites; the concentration increases from 25 to 28 March and reaches a maximum on 30M in Barcelona and Bellaterra and on 31M in Tarragona, which is followed by a sudden decrease on 1 and 2 April when the pollination event vanishes. On the other hand, the model does not capture the measurements in the other XAC sites (Girona, Lleida, Manresa and Roquetes; not shown). In some cases, the pine trees did not pollinate during the event but two weeks later (e.g., Manresa) or the week before (e.g., Roquetes) (see http://lap.uab.cat/aerobiologia); in other places the distribution of trees imposed in the model (Figure 1a) might not be accurate enough (e.g., Girona). Even in a relatively small geographical region, the behaviour





**Table 6. Statistics (Pearson correlation coefficient *r*, root mean square error *rmse*, and bias, *bias*) of the model surface pollen concentration daily mean and meteorology hourly mean for 9 km domain (D02) vs. measurements calculated over the period 25 March to 2 April of the event. Measurement sites are detailed in Table 1.**

| XAC/XEMA site | Pollen conc. (Pollen m⁻³) | Temperature (°C) | Wind Speed (m s⁻¹) | Wind Direction (°) | Solar Radiation (W m⁻²) | Relative Humidity (%) |
|---|---|---|---|---|---|---|
| | | | ***r*** | | | |
| Barcelona | 0,5 | 0,9 | 0,3 | 0,8 | 1,0 | 0,7 |
| Bellaterra/Sant Cugat | 0,9 | 0,9 | 0,4 | 0,6 | 1,0 | 0,7 |
| Girona | 0,2 | 0,9 | 0,5 | 0,5 | 1,0 | 0,8 |
| Lleida | 0,4 | 0,9 | no data | no data | no data | 0,9 |
| Manresa | -0,6 | 0,9 | 0,3 | 0,4 | 1,0 | 0,8 |
| Roquetes/Aldover | -0,3 | 0,9 | no data | no data | 0,9 | 0,9 |
| Tarragona | 0,6 | 1,0 | 0,5 | 0,1 | 1,0 | 0,7 |
| | | | ***rmse*** | | | |
| Barcelona | 370,0 | 2,8 | 1,8 | 65,6 | 71,5 | 16,2 |
| Bellaterra/Sant Cugat | 477,1 | 2,2 | 1,2 | 70,2 | 74,7 | 17,3 |
| Girona | 450,1 | 2,8 | 1,4 | 84,6 | 85,8 | 13,3 |
| Lleida | 102,4 | 1,8 | no data | no data | no data | 7,2 |
| Manresa | 1287,8 | 2,2 | 3,1 | 96,5 | 96,0 | 17,9 |
| Roquetes/Aldover | 473,7 | 4,1 | no data | no data | 108,3 | 16,4 |
| Tarragona | 231,7 | 2,2 | 2,5 | 71,5 | 54,3 | 10,7 |
| | | | ***bias*** | | | |
| Barcelona | 111,3 | 0,3 | -0,2 | -13,0 | 8,3 | 0,4 |
| Bellaterra/Sant Cugat | -366,0 | 1,1 | 0,7 | 8,7 | 19,3 | -12,0 |
| Girona | -337,3 | 0,2 | 1,0 | -13,9 | 9,0 | -5,3 |
| Lleida | -39,3 | -1,1 | no data | no data | no data | -0,3 |
| Manresa | 931,2 | 0,3 | 2,8 | -32,2 | 28,4 | -11,0 |
| Roquetes/Aldover | -252,7 | -3,8 | no data | no data | 23,6 | 15,5 |
| Tarragona | -80,0 | -1,7 | 1,6 | -4,9 | -12,0 | 0,8 |

of the pine trees is significantly heterogeneous due to the types of trees present, different micro-climates, and meteorological

5  conditions. It is clear that more efforts are needed to develop accurate phenology functions and detailed maps to improve our

current results, but the skills of the model over the Barcelona site are good enough to study the vertical structure of pollen

there.

**5.2. Surface concentration in Barcelona: observations and modelling**

In this section, we discuss the hourly resolution results of the surface concentration at the Barcelona XAC site where specific

10  high-temporal resolution measurements were available for the period 27M to 31M. Figure 9 shows the time series of the hourly

*Pinus* surface concentration of the model at 9, 3 and 1 km resolution (colour lines) and the observations (black line) for the







Figure 7. Comparison of the model and observations at (a) Barcelona and (b) Bellaterra/Sant Cugat XEMA sites of 2-meter air temperature, 10-meter wind speed and direction, incoming shortwave radiation at surface, and 2-meter relative humidity, during the 5 days of the event. Color lines represent model results at 9, 3 and 1 km resolution.

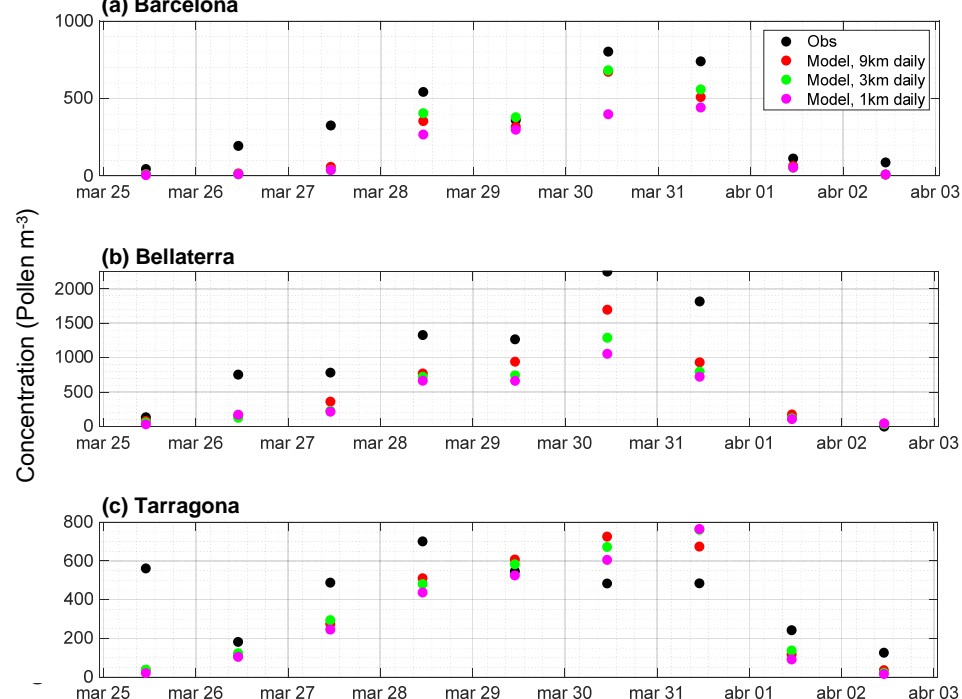

**Figure 8. Daily average *Pinus* surface number concentration at (a) Barcelona, (b) Bellaterra and (c) Tarragona XAC sites for the period 25 March to 2 April 2015. Colour dots represent model results at 9, 3 and 1 km resolution.**

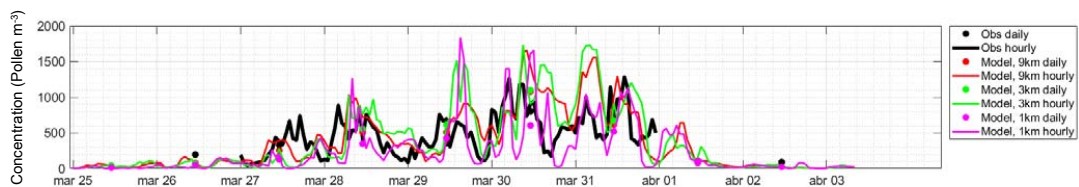

**Figure 9. Comparison of the model forecast and the observations of *Pinus* surface number concentration during the 5 days of the event for the Barcelona site.**

Barcelona site. Complementary, we plot the daily averages that extend beyond the period where high-temporal resolution measurements are available. As discussed in the previous section, the model has a delay in the rise of pollen concentration observed from 25M to 27M. During that period, although the model emits pollen grains (see Figure 6), the concentrations in the air are strongly underestimated. The model starts to reproduce the observations the morning of 28M, and the agreement with the observations is maintained for the rest of the event. The three model resolutions follow the trend of the measurements reasonably well. The 1 km run tends to simulate sudden rises and falls in the concentrations compared with the 9 and 3 km, which are able to maintain some background concentrations in the air. The model has significant skills to reproduce the temporal variability of the observations with sudden peaks on 30M and 31M or sustained high concentrations the midday of 28M or afternoon of 31M. Some model underestimations can be explained by the biases in the winds. For example, the strong underestimation during the first hours of 28M is attributed to the underestimated wind peak observed in Figure 7a, or the underestimation during 29M in the morning is associated by a wrong wind direction (northerly winds in the model versus easterly winds in the observations). In this sense, the 1 km simulation is more sensitive to the errors in the winds than the upper domains. Note the need of some wind intensity in the model to capture the concentrations; under weak conditions the model





simulates extremely low concentrations pointing to the need of a minimum mobilisation term in the emission scheme as suggested by other studies (e.g., Sofiev et al., 2013).

To compare the skills of the model at different horizontal resolution, Table 7 presents the day-by-day statistics for the hourly *Pinus* number concentration at the Barcelona site. None of the domain resolutions is doing significantly better than the others,

although, overall, the domain with the best statistical indicators (lowest *rmse* and *bias* and highest *r*) is the 9 km domain. The start of the event is better captured by the 9 km but the 1 km results improve significantly afterwards resulting in lower biases. The correlations are low in most cases (around 0.2 - 0.3) highlighting the complexity to reproduce the hourly variability in pollen models.

**Table 7. Day-by-day statistics (root means square error, *rmse*, Pearson correlation coefficient, *r*, and bias, *bias*) of the hourly model**

**(*Basecase*) *Pinus* surface number concentration and measurements calculated over the 5 days of the event at the Barcelona site.**

| | | 27M | 28M | 29M | 30M | 31M |
|---|---|---|---|---|---|---|
| | **Conc. Obs. (Pollen m$^{-3}$)** | 286 | 433 | 387 | 700 | 668 |
| **9 km (D02)** | **Conc. Model (Pollen m$^{-3}$)** | 214 | 532 | 459 | 941 | 882 |
| | *rmse* **(Pollen m$^{-3}$)** | 205,7 | 283,8 | 329,5 | 534,0 | 488,9 |
| | *r* | 0,3 | 0,4 | 0,1 | -0,1 | 0,2 |
| | *bias* **(Pollen m$^{-3}$)** | -71,2 | 98,9 | 72,5 | 240,9 | 213,5 |
| **3 km (D03)** | **Conc. Model (Pollen m$^{-3}$)** | 115 | 529 | 599 | 876 | 1006 |
| | *rmse* **(Pollen m$^{-3}$)** | 281,4 | 351,4 | 469,0 | 592,8 | 650,9 |
| | *r* | -0,1 | 0,1 | 0,3 | -0,3 | -0,1 |
| | *bias* **(Pollen m$^{-3}$)** | -170,2 | 96,3 | 212,0 | 176,4 | 337,3 |
| **1 km (D04)** | **Conc. Model (Pollen m$^{-3}$)** | 90 | 337 | 405 | 555 | 594 |
| | *rmse* **(Pollen m$^{-3}$)** | 295,6 | 342,3 | 421,2 | 555,8 | 419,1 |
| | *r* | -0,2 | 0,1 | 0,3 | 0,2 | 0,1 |
| | *bias* **(Pollen m$^{-3}$)** | -195,6 | -95,9 | 18,2 | -144,8 | -73,9 |

### 5.3. Concentration in the column in Barcelona: observations and modelling

The study of the mechanisms responsible for the pollen transport and the skill analysis of the model to predict the vertical distribution of *Pinus* grains are performed in terms of two vertically integrated statistical indicators, namely the fractional bias (*FB*), and the Pearson correlation coefficient ($r$). *FB* and $r$ are both calculated for the number concentration. For each vertical

1-hour profile, the vertical extension considered starts at the lowest pair of simultaneously available model and observed values (fixed at 225 m, the height of the first valid lidar measurement) and ends at the pollen top height, $h_{pol}$, taken from Sicard et al. (2016a). Both the *Basecase* (100% deposition, 100% sedimentation) and the *Enhanced* (50% deposition, 50% sedimentation) simulations are considered, both at the three domain resolutions of 9, 3 and 1 km.

Before entering in the analysis of the results, we present the latitudinal and longitudinal cross-sections of the model *Pinus*

number concentration at the coordinates of Barcelona lidar site (2.112ºE, 41.389ºN; see Figure 6 red dashed lines) for the 1 km resolution on 30M at 12 UTC (Figure 10). More vertical structure is observed in the longitudinal cross section compared to the latitudinal one where the orography is more pronounced (Pyrenees). In agreement with the emission scheme, most of the pollen dispersion occurs downwind of the second mountain range from the shore (latitudinal cross section). Above the lidar site, the model predicts a thick *Pinus* pollen layer up to ~0.8 km and then a decrease up to ~1.3 km. As it will be shown




later, the model vertical structure is not always in agreement with the measurement and presents more variability with increased horizontal resolution, particularly from noon onwards.

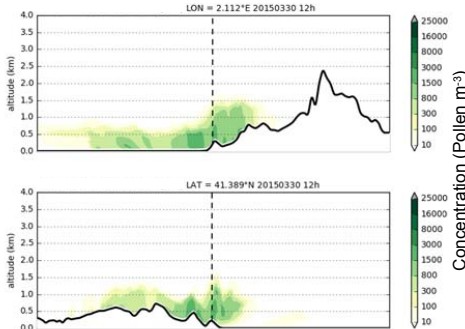

**Figure 10. Vertical cross section of the *Pinus* number concentration (Pollen m⁻³) simulated at 1 km resolution (D04) at 12UTC on 30M. The horizontal latitudinal (top panel) and longitudinal (bottom panel) cross sections are reported with red, dash lines in the corresponding map of Figure 6.**

In Table 8 we report the daily average of $FB$ and $r$ for all five days of the event for both simulations at all three domain resolutions (9, 3 and 1 km). The fractional bias and the correlation coefficient as a function of domain resolution are shown for the two simulations in Figure 11. Figure 12 shows for each of the five days of the event the results of the *Basecase* simulation at 9 km resolution (the resolution resulting in the lowest statistical error at the surface, see Section 5.2): daily mean vertical profiles of *Pinus* number concentration simulated by the model and measured by the lidar (top panels), the temporal evolution of the hourly $FB$ and $r$ (middle panels), and the plot of hourly $FB$ vs. $r$ (bottom panels). According to Table 8, the model significantly underestimates the *Pinus* number concentration in the column on the first (27M, $FB < -60\%$) and last (31M, $FB < -30\%$) day of the event, a result similar to what is found at the surface (Section 5.2). $FB$ is in general smaller (in absolute value) and closer to zero on 28M, 29M and 30M than on 27M and 31M. This is partly due to the definition of the phenology function which is fitted to the surface concentration regardless of the vertical distribution. While the *Pinus* daily mean number concentration increases 50 % between 27M and 28M, the pollen AOD remains constant between both days (Sicard et al., 2016a). In these conditions our phenology function will yield an underestimation of the vertical distribution of *Pinus* pollen in the column on 27M, as it is observed. The same occurs on 31M: while the *Pinus* daily mean number concentration decreases between 30M and 31M, the pollen AOD remains constant between both days, hence the underestimation of the vertical distribution of *Pinus* pollen in the column observed on 31M. This result suggests that the phenology function, fitted to the surface concentration, works relatively well to also reproduce the quantity of pollen transported in the column, at least during the three most intense days of the event. It is important to point out the difference between the daily $FB$ (Table 8) and the daily mean vertical profiles (top plots in Figure 12) to avoid misinterpretations: the first one is vertically-integrated while the second one is not; the first one represents a relative difference while the second one represents an absolute difference. Nighttime and daytime hourly $FB$ weights the same in the calculation of the daily $FB$, although sometimes the nighttime $FB$ is calculated only over a few tens of meters and the daytime $FB$ is calculated over several hundreds of meters. Contrarily, the average of a near-zero nighttime profile and a daytime profile with non-zero concentrations will reflect essentially the shape of the daytime profile (qualitatively, not quantitatively). These differences explain why on 30M, e.g., $FB$ is negative for the *Basecase* at 9 km resolution ($-11.3\%$, Table 8) while the mean profile is greater than the observation (Figure 12). If one looks at the evolution of $FB$ with time, although the daytime $FB$ is positive, one sees that $FB$ is strongly negative ($< -100\%$) during the night hours, producing a daily mean slightly negative ($-11.3\%$).

The daily correlation coefficient is higher than 0.41 in all cases (Table 8) and varies between 0.41 and 0.93. At 9 km resolution (the resolution chosen for Figure 12) $r$ varies between 0.41 and 0.90. The middle plots of Figure 12 show that each day many





values of $r$ approach very closely 1.00, especially on 28M, 29M and 30M. On these three days, the minimum values of $r$ are obtained for the finest resolution of 1 km. For the same period, the daily mean correlation coefficient is greater than 0.83 and 0.70 for 9 and 3 km resolutions, respectively. This result suggests that the model reproduces quite well the shape of the *Pinus* number concentration vertical distribution, at least during the three most intense days of the event ($r > 0.70$). The bottom

plots of Figure 12 summarize well the score of the model as far as vertically-integrated $FB$ and $r$ are concerned: on 28M, 29M and 30M the pair ($FB$, $r$) is very close to the ideal value, (0, 1), indicated by a grey circle. However these plots also show the large variability of the hourly values of $FB$ varying from negative (underestimation, nighttime) and positive (overestimation, daytime).

**Table 8. Daily statistics (fractional bias, $FB$, and correlation coefficient, $r$) of the modelled vertical distribution concentration vs.**
**lidar-derived vertical distribution for the two simulations, each one in the three domain resolutions.**

|  |  | *Basecase* simulations (100% deposition, 100% sedimentation) | | | *Enhanced* simulations (50% deposition, 50% sedimentation) | | |
|---|---|---|---|---|---|---|---|
|  |  | 9 km (D02) | 3 km (D03) | 1 km (D04) | 9 km (D02) | 3 km (D03) | 1 km (D04) |
| 27M | $FB$ (%) | -82.3 | -87.7 | -94.6 | -60.8 | -64.0 | -69.4 |
|  | $r$ | 0.53 | 0.51 | 0.52 | 0.53 | 0.49 | 0.51 |
| 28M | $FB$ (%) | +28.3 | +26.9 | -47.6 | +47.9 | +46.6 | -31.1 |
|  | $r$ | 0.83 | 0.92 | 0.77 | 0.83 | 0.93 | 0.73 |
| 29M | $FB$ (%) | -8.2 | -7.8 | -48.9 | +19.5 | +13.0 | -23.3 |
|  | $r$ | 0.84 | 0.71 | 0.52 | 0.83 | 0.70 | 0.48 |
| 30M | $FB$ (%) | -11.3 | -42.0 | -113.7 | -1.9 | -34.8 | -108.9 |
|  | $r$ | 0.90 | 0.90 | 0.77 | 0.90 | 0.90 | 0.76 |
| 31M | $FB$ (%) | -38.3 | -57.1 | -85.9 | -30.4 | -46.8 | -80.9 |
|  | $r$ | 0.43 | 0.47 | 0.48 | 0.41 | 0.47 | 0.44 |

**Figure 11. Daily (a) fractional bias vs. domain resolution and (b) correlation coefficient vs. domain resolution for the *Basecase* (solid lines) and *Enhanced* (dash lines) simulations.**

A lot of variability of $FB$ is also observed as a function of the domain resolution considered for the simulation (Figure 11a).
In almost all cases, the simulations at 1 km resolution give lower $FB$ (stronger underestimation) than at 9 and 3 km resolution. Except on 30M, day for which the 3 km resolution gives a significantly lower $FB$ than the 9 km one, the 9 and 3 km resolution yields similar $FB$. The correlation coefficient can also vary significantly from one resolution to another: a difference of up to 0.35 (*Enhanced* simulation on 29M) is observed between 9 ($r = 0.83$) and 1 km ($r = 0.48$) resolutions (Figure 11b). Differences between 9 and 3 km resolutions are not higher than 0.13 (both simulations on 29M). On 30M and independently
of the simulation version, the correlation coefficient is remarkably constant and equal to 0.90 for the two resolutions of 9 and





km. In agreement with the surface results (Section 5.1 and 5.2), errors in the meteorology have a major impact in the dispersion of pollen at higher model resolution. While the structure of the plumes downwind emission sources are more well defined at 1 km resolution, small errors in the wind speed or direction result in larger errors at such high mesoscale resolution that are smoothed at coarser resolutions due to the numerical diffusion. This is a classical problem when mesoscale models

approach the 1 km horizontal resolution. From the above discussion we conclude that the 9 and 3 km resolutions are probably the most suitable to minimize bias and maximize correlation between *Pinus* pollen forecast and observations in the column. The 9 km resolution is chosen for plotting the hourly and daily mean variations shown in Figure 12.

The effect of the deposition and sedimentation on the quantity of pollen grains transported vertically is studied by means of the two simulations defined in Section 3.3: *Basecase* (100% deposition, 100% sedimentation) and *Enhanced* (50% deposition,

50% sedimentation) simulations. The results are reported in Table 8, Figure 11 and the top plots of Figure 12. The study on the sedimentation is motivated by the high sedimentation velocity of large *Pinus* grains ($3 - 4$ cm·s$^{-1}$; Jackson and Lyford, 1999). As an example, particles of mineral dust of diameter smaller than 10 μm have a sedimentation velocity lower than 1 cm·s$^{-1}$ (Li and Osada, 2007). Pollen sedimentation velocity is in partly controlled by the grain size, density and shape which are highly variable. Consequently, the sedimentation velocity may be one of the main factors regulating the pollen vertical

dispersion. In the case of *Pinus* pollen, the sacci, or inflated air bladders on each side of the grain, play an additional role in such a mechanism, as they act as aids for aerial dispersal (Wodehouse, 1935; Proctor et al., 1996; Schwendemann et al., 2007). As seen in Figure 11a, the combined effect of deposition/sedimentation on *FB* is quite notable. In the cases for which the *Basecase* simulations result in an underestimation (negative daily *FB*, see Table 8), the *Enhanced* simulations, as expected, always reduce this underestimation and thus improve the model score. Contrarily, when the *Basecase* gives a positive daily

*FB*, the *Enhanced* simulation increases this value and thus worsen the overestimation. In terms of fractional bias, the increase of daily *FB* when the deposition/sedimentation is set to half of its nominal value varies between 4.8% (30M, 1 km) and 27.7% (29M, 9 km). In average over the five days of the event, the mean increase in daily *FB* decreases with decreasing domain resolution: Δ*FB* is 17.2, 16.3 and 15.4 for 9, 3 and 1 km resolutions, respectively. The finest the resolution, the less sensitive to deposition/sedimentation the simulations. The effect of the reduction of deposition/sedimentation is also visible on the top

plots of Figure 12 in which the results of *Enhanced* simulations are also reported. In all cases the concentration increase due to the reduction of deposition/sedimentation is significant. As expected, larger differences are observed in the lowermost layers. The daily relative increase in concentration (vertically-integrated up to 2 km height, see top plots of Figure 12) due to a 50% decrease in deposition/sedimentation is 16.1, 14.1, 16.5, 13.0 and 9.7% on 27M, 28M, 29M, 30M and 31M, respectively. The effect of the deposition/sedimentation reduction on the concentration is not linear. The deposition/sedimentation has a

small effect on the capability of the model to reproduce the shape of the *Pinus* pollen vertical distribution (Figure 11b): from *Basecase* to *Enhanced* simulations the least significant figure of the correlation coefficient does not change by more than ±0.04. If we restrict to the three most intense days of the event and to the 9 and 3 km resolution simulations |Δ*r*| ≤ 0.01. In conclusion of this analysis, reducing deposition/sedimentation has virtually no impact on the way the model reproduces the shape of the vertical distribution of the pollen number concentration and suggests that other processes are more relevant to

explain the vertical structures observed.

To have a closer look to the performance of the model in the column with respect to time, we plot in Figures 13, 14 and 15 the hourly evolution of the profiles of *Pinus* number concentration for the *Basecase* simulation at all three domain resolutions and for the observations on 28M, 29M and 30M, respectively. On 28M, 29M and 30M, the three most intense days of the event, a "bi-modal" diurnal cycle of the fractional bias is visible in Figure 12 (middle plot) underestimation during nighttime and two

peaks of overestimation at 6-10UTC and at 15-16UTC. The morning peak is probably due to an excessive mobilization of pollen during nighttime associated to a possible lack of ventilation during the first hours of the day. As seen in Section 5.1 and in Figure 7 in particular, the model has some systematic error inducing overestimation of nocturnal winds and underestimation of the increase of winds during the morning. The afternoon peak is likely due to the fact that pollen stays aloft longer than it





does in reality or an excessive emission flux during the previous hours upwind. This behaviour of the model is present in all Figures 13, 14 and 15. According to the lidar, the pollen layer grows between 9-14UTC following a typical development of the convective boundary layer. During that period the top of the layer is well reproduced by the model with excessive mobilisation of pollen grains depending on the horizontal resolution and hour. Note the good agreement in some specific

profiles, see e.g. on 30M at 12-13UTC 3 (Figure 15). The sink of the layer is observed between 14-15UTC on 28M and 29M (the model predicts it 1 hour later) and between 13-14UTC on 30M (the model predicts it 2 hours later). The delay of the model in predicting the pollen layer drop in the afternoon might be linked to the fact that the model delays the decay of the convective boundary layer. In the afternoon and especially for the 1 km resolution the model predicts a strong and steep decrease in concentration towards the surface. The same result is observed with the surface concentration in Section 5.2 where the

simulation at 1 km resolution shows a decrease in concentration (which leads to an underestimation of the model, see Figure 9) followed by sudden peaks that correlate well with the wind speed. Again, it shows the higher sensitivity to wind speed and direction of the 1 km simulation compared to the 3 and 9 km simulations. Also, more structures are visible at 1 km resolution than at the other resolutions. The finer the resolution, the more vertical structures. However the structures visible at the 1 km resolution are not always reproducing reality, see e.g. on 29M at 14UTC (Figure 14).

The Hirst observations are much more variable than the model concentration and the meteorology. Although following a general trend, the Hirst concentrations oscillate up and down most of the time and along the whole day (see the black circles in the hourly plots of Figures 13, 14 and 15, and the black line in Figure 9), while the model concentration usually steadily increases in the morning and decreases in the afternoon after 15-17UTC. Such a difference between model and observations is not visible on the meteorological variables (Figure 7). In fact, for some variables, the opposite happens: e.g. the model wind

speed is more variable than the observation. The high variability of the observed *Pinus* concentration, independently of the meteorology, emphasizes again the difficulty to predict airborne pollen grain concentration and the need to develop emission schemes especially designed for bioaerosols, the emission of which is much more complex than that of the rest of natural particles.

Finally, by looking at Figures 13, 14 and 15, the general underestimation of the model during nighttime mentioned earlier is

not obvious. If we look back to Figure 5c, one sees differences in the nighttime vertical structure. The quicklook of the volume depolarization ratio is either strong (yellow/orange, pollen in large amount), or small (light green, pollen in small amount) or very small (dark green/black, molecular level). Layers with small amount of pollen seem to detach from the surface pollen layer in the afternoon on 27M, 28M, 29M and 31M. They are likely residual pollen layers made probably of smaller pollen grains, not necessarily *Pinus* pollen and not necessarily emitted in the geographical distribution considered, that enter the free

troposphere. The fact that they show up in the afternoon and are coupled to the main pollen layer suggests that they might be made of local pollen. In such a case, their formation is likely similar to that of the residual layer that usually settles above the nocturnal stable boundary layer. Further research, out of the scope of this paper, is needed for clarifying these aspects about the origin of these residual pollen layers. We will take two nighttime examples: 28M at 03UTC (no residual pollen layer, $h_{pol}$ = 0.45 km, $FB$ = -36.0 %) and 30M at 03UTC (presence of a residual pollen layer, $h_{pol}$ = 1.74 km, $FB$ = -129.8 %). We refer

to Sicard et al. (2016a) for the definition of the pollen layer height, $h_{pol}$: $h_{pol}$ was calculated as the height at which the $\beta_{pol}$ < 0.055 Mm$^{-1}$sr$^{-1}$. Figure 16 shows these two examples in terms of quicklooks of the range square-corrected signal and hourly profiles of the *Pinus* number concentration. Unlike in Figures 13, 14 and 15, the *Pinus* number concentration in Figure 16 is plotted in a logarithmic scale going from 10$^{-10}$ to 1000 Pollen m$^{-3}$. On 28M the surface layer is detected very close to the ground ($h_{pol}$ = 0.45 km) and, as expected from the comparison (Figure 16a) of the model (red) and the lidar (black) the model slightly

underestimates the observation ($FB$ = -36.0 %). This feature can not be well appreciated in Figure 13 because the x-scale is linear and goes from 0 to 1000 Pollen m$^{-3}$. On 30M, a clear surface pollen layer is seen up to 0.40 km, but then the end of the residual pollen layer is detected at 1.74 km with the employed method. Between 0.40 and 1.74 km pollen is detected in very small amount because the total backscatter coefficient, although small, is not zero and the particle depolarization ratio is





slightly larger than the reference value taken for non-depolarizing particles (0.03, see Sicard et al. (2016a)). From Figure 16b, one sees that the model does relatively well up to $0.40 - 0.50$ km but then strongly drops to near-to-zero values ($\sim 10^{-7} - 10^{-5}$ Pollen m$^{-3}$ at $h_{pol}$). Despite the small amount of pollen observed between 0.40 and 1.74 km, since $FB$ is a relative difference, the near-to-zero values of the model produce large negative fractional biases ($< -100\%$). After going through the nighttime

5    profiles one by one, we noticed that the model profiles systematically drop very quickly above the surface pollen layer and, in case a residual pollen layer was observed, the latter was not reproduced by the model. However, this does not mean the model does it wrong since we are not fully sure that the pollen observed in these residual pollen layers is *Pinus* pollen. Worth mentioning that the uncertainties of the model processes and numerical schemes implemented in Eulerian models like the one used in our study are major limitations to reproduce properly such low concentrations in residual layers and one could argue

10   that the model result below such thresholds approach the numerical noise. This paragraph is only intended to highlight one of the limitations of our methodology related to the presence of residual pollen layers. As said earlier, this issue calls for further research.



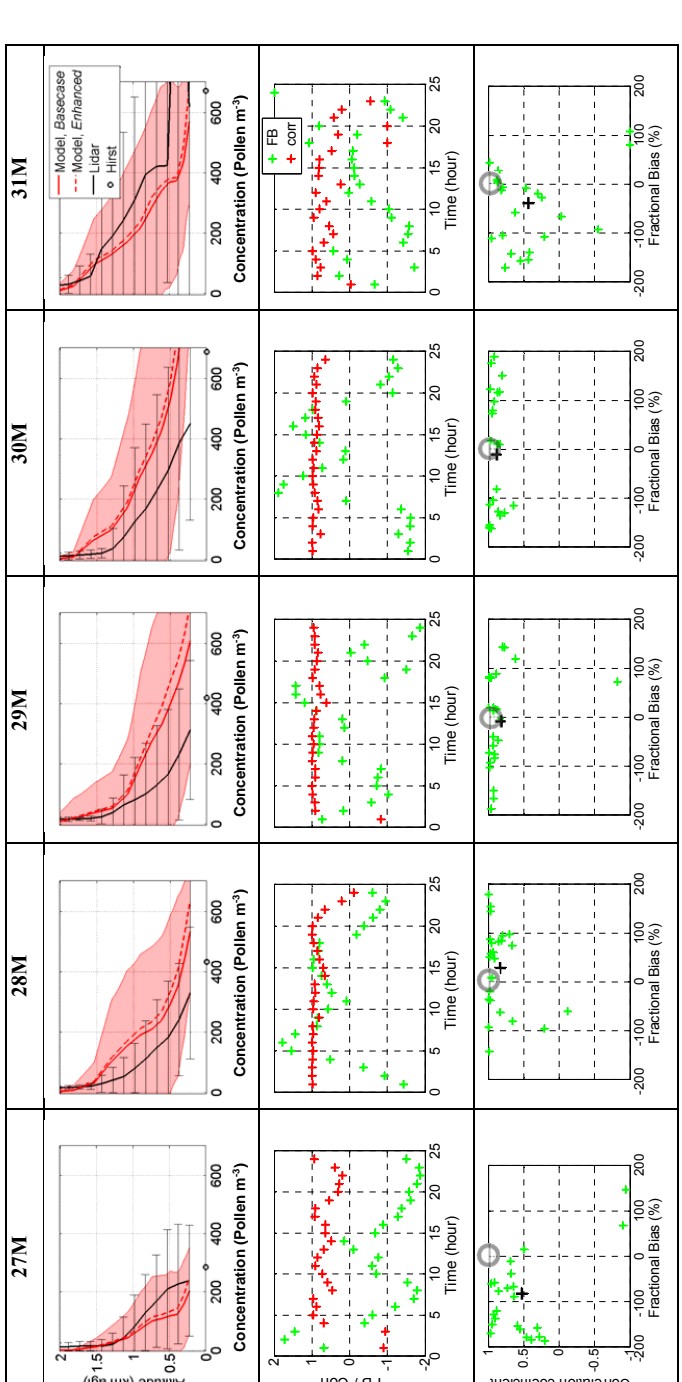

**Figure 12.** For each of the five days of the event (columns) (top) daily mean modelled *vs.* observed vertical profile of the *Pinus* number concentration; (middle) 1-24 hour time evolution of the fractional bias, *FB*, and correlation coefficient, *r*; (bottom) *r vs. FB*. In the top plots: the daily mean Hirst surface concentration is reported as a circle at the ground level for reference; the model and lidar standard deviations are reported as shaded area and horizontal bars, respectively. In the bottom plots: the ideal (*FB*, *r*) values, (0, 1) are indicated by a grey circle. The black crosses on the bottom plot are the daily mean value. The simulation considered is the *Basecase* with the domain resolution of 9 km (D02). In the top plots, the daily mean vertical profile of the *Enhanced* simulation is also reported for comparison.



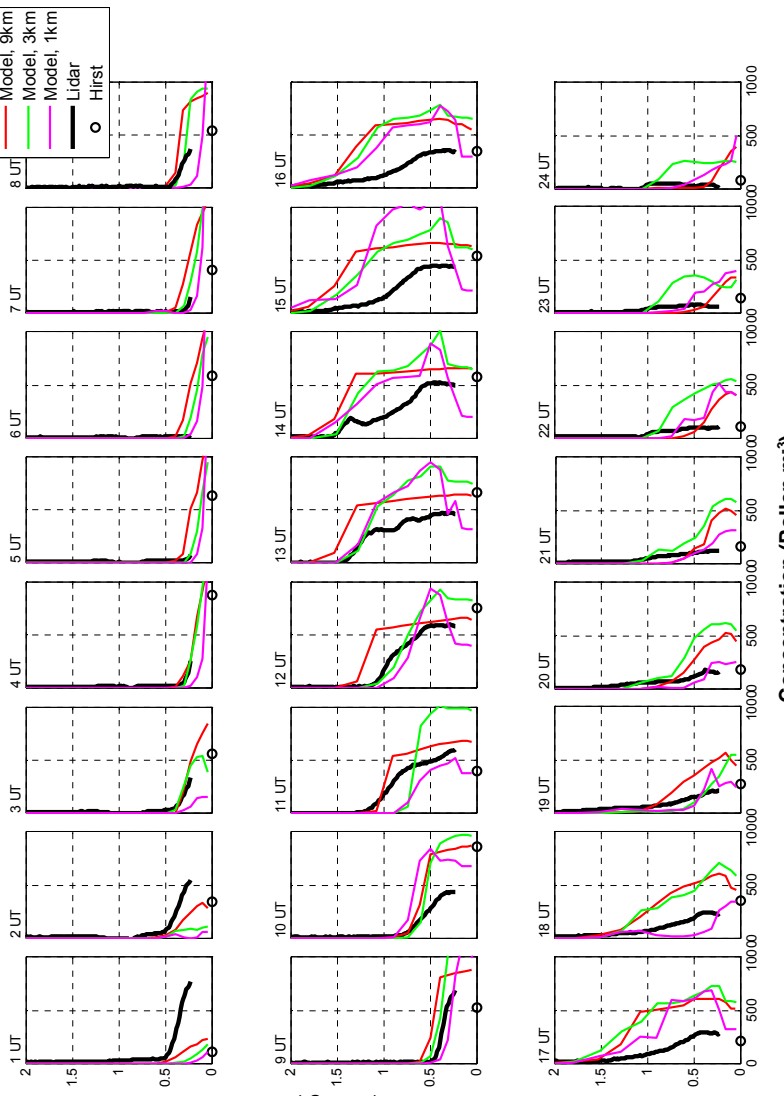

**Figure 13. Hourly evolution of the vertical profile of *Pinus* number concentration on 28M, forecast by the *Basecase* simulation for the three domain resolutions. Hourly lidar–derived vertical profiles are reported in black. Hourly Hirst surface concentrations are reported as a black circle at ground level for reference.**





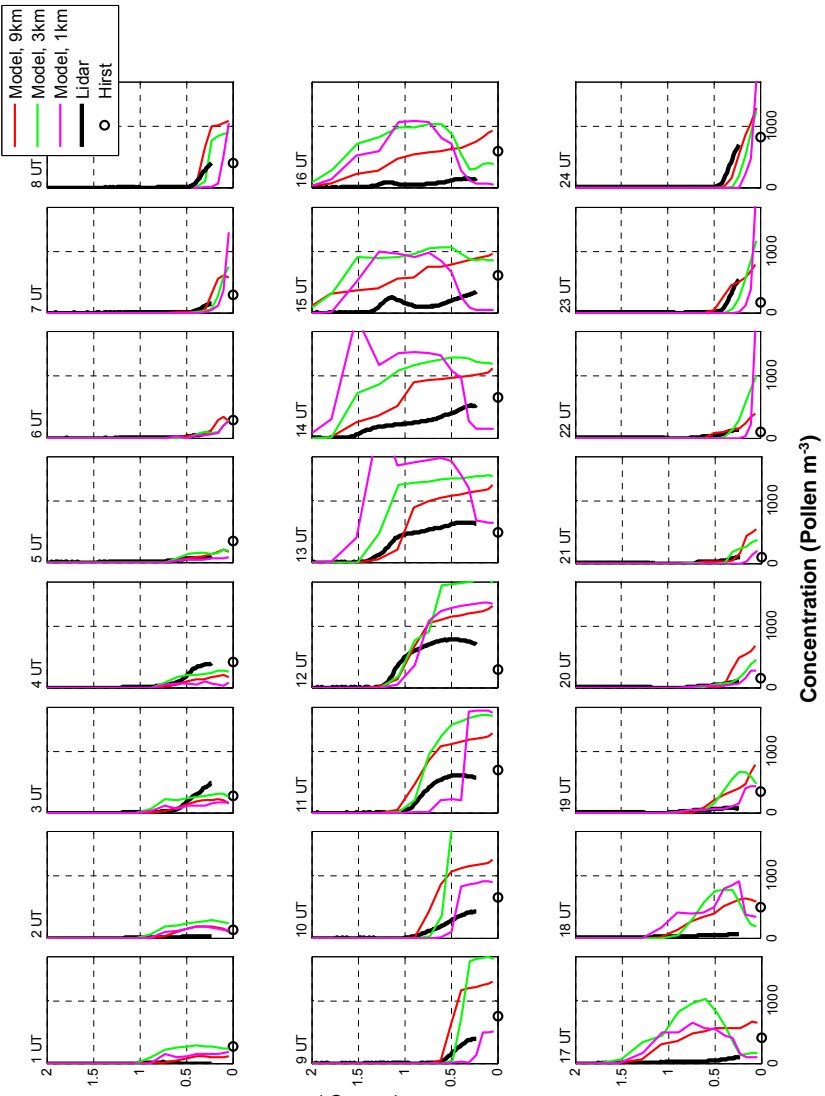

**Figure 14. Idem as Figure 13 for 29M.**



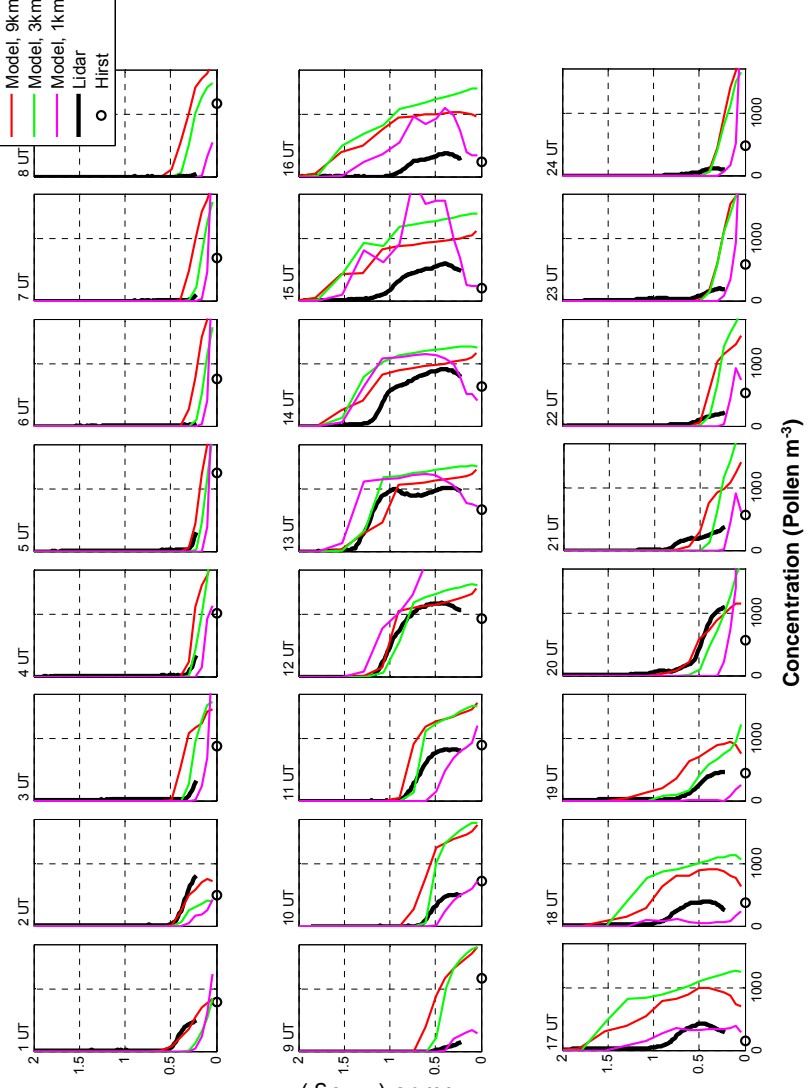

**Figure 15. Idem as Figure 13 for 30M.**





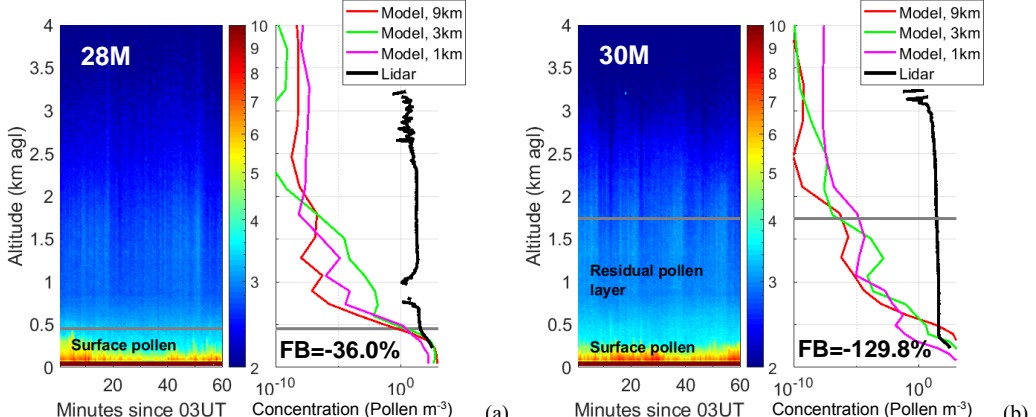

**Figure 16. Quicklooks of the range square-corrected signal and hourly profiles of the *Pinus* number concentration on (a) 28M at 03UTC and (b) 30M at 03UTC. $h_{pol}$ is indicated by a grey horizontal line. *FB* is reported for the resolution of 9 km.**

## 6. Conclusions

This paper combines pollen concentration surface (Hirst) and columnar (lidar) measurements as well as an atmospheric transport model with a simplified pollen module especially implemented for this study to improve our understanding of pollen vertical dispersion, distribution and mixing in the atmospheric column. The pollen type under study is *Pinus* and the event of interest is a 5-day pollination event which occurred in the region of Barcelona, NE Spain, during 27 – 31 March, 2015. Prior to the analysis, conversion formulas are given to convert lidar-derived total backscatter coefficient and model-derived *Pinus* pollen mass concentration into *Pinus* pollen number concentration, the magnitude measured at the surface. The most sensitive conversion being the one of the backscatter coefficient, which is based on the specific extinction cross-section, a critical parameter virtually unknown, a sensitivity analysis is carried out on the latter. Its estimation is made by minimizing the sum of squared residuals between the surface concentration and the first lidar measurement taken as a proxy of what it would be at ground level. In the conditions of our case study the specific extinction cross-section oscillates between 0.78 and 1.67 m²g⁻¹. To our knowledge, it is the first time a relationship between optical and mass properties of atmospheric pollen is quantified in ambient conditions.

Nested numerical simulations at 9, 3 and 1 km horizontal resolution are deeply analyzed, as well as the effects of sedimentation and dry deposition. The model used is the Multiscale Online Nonhydrostatic AtmospheRe CHemistry model (MONARCH) developed at BSC. *Pinus* pollen grains are represented as a spherical particle with hygroscopic growth factors following Griffiths et al. (2012) affecting the density of the particle while keeping its size constant. The emission scheme is based on the concepts of the parameterization of Helbig et al. (2004). One of its most critical parts is the phenology function which in our work is constrained by the observed pollen counts by fitting a Gaussian function to the evolution of the pollen concentration during the period 1 March – 30 April, 2015. The spatial distribution of pine trees, obtained from the Cartography of habitats of Catalonia (Carreras et al., 2015) and the inventory of the Barcelona's City Council, has been remapped to a 1 km resolution dataset and then combined with the pine tree density reported by the Forest Inventory of Catalonia (Gracia et al., 2000-2004). The representativeness of the model results is assessed by comparing them with hourly meteorological and daily aerobiological observations in several points of Catalonia. In general, the statistics indicates a good agreement for temperature, irradiance and relative humidity in most of the stations (correlations above 0.9 and low bias), while higher errors are observed in the winds. A better statistic is observed at inland sites compared to coastal sites. The good agreement of the model meteorology with the observations in the horizontal scale is a mandatory requirement for the assumptions made to "calibrate" the lidar with the Hirst sampler (which are 4.4 km apart). At the Barcelona site where the lidar is located, an improvement is detected in most



meteorological variables with the increase of resolution. The model reproduces the daily cycle of the temperature and relative humidity; more disagreement is seen in the wind speed, and the different resolutions, although consistent one to another, show larger variability compared with the other variables. The model overestimates the calm winds observed during nighttime and it tends to underestimate the morning peak. This results in lower relative humidity and higher temperatures during the calm

periods compared with the observations. When combining both analysis of hourly surface and column concentration vs. model the following results are found:

- The three model resolutions follow the evolution of the measurements in general. The 1 km run tends to simulate sudden rises and falls in the concentrations compared with the 9 and 3 km, which are able to maintain some background concentrations in the air. Although the simulation at 1 km may improve the model score in places with complex

topography, the meteorological errors in winds have larger impacts in such a high resolution configuration. In this sense, the combination of the three resolutions provides complementary information to advance our understanding in key driving processes, but 9 or 3 km simulations might be preferred for specific forecasting applications.

- The largest discrepancies between measured surface (Hirst) and column (lidar) concentrations occur during nighttime: no pollen is detected in the column whereas it is present at the surface. This is likely due to the limitation of the lidar to

measure below 225m where most of the stable boundary layer resides. Simulated profiles at different resolutions show large variability throughout the event of study. During nighttime the model tends to overestimate the amount of pollen available in the stable boundary layer compared with the Hirst concentration, but matches its top when it is detected by the lidar. Such overestimation may be attributed to systematic errors in the wind speed during midnight. This points out that the wind is the main driver of the nighttime/early morning pollen activity.

- A 50% decrease of the sedimentation/deposition parameter increases the daily column concentration 10-17 %. Decreasing the deposition/sedimentation in the model is not enough to significantly change the results. Both parameters only have a limited impact on the vertical concentration suggesting that other processes are more relevant to reproduce the measurements.

- From our model results, the vertical structure of the pollen is mainly controlled by the vertical mixing within the boundary

layer and the sensitivity of the emission scheme to winds. In general terms, the model matches the depth of the pollen layer during stable conditions and the growth of the convective boundary layer, but systematic biases are detected in the second half of the day with persistent overestimation within the convective boundary layer. During the latter period, lidar profiles do not extend up to the top of the boundary layer, probably because of gravity effects that are not well represented by the model.

- Finally, during nighttime, lidar observations reveal the presence of residual pollen layers that are not reproduced by the model in its present configuration: no clear exchanges of pollen grains between the boundary layer and the free troposphere are identified in the simulations. Given the uncertainty related to the origin and formation of these residual pollen layers and the type of pollen they contained no conclusion about the model score can be drawn at this stage. This result calls for further research.

The results of the study emphasizes the tremendous importance of the completeness of the tree spatial distribution, density and type: even in a relatively small geographical region, the behaviour of the pine trees is significantly heterogeneous due to the presence of different pine species, different micro-climates, and meteorological conditions. The assumption made in the model that the availability of pollen grains in the pine trees (*P. halepensis*, *P. pinea*) of the region is the same during the pollination event is far from being close to reality. A clear outcome of this study is the need for more research in the development of

phenology functions for bioaerosols, which, unlike the rest of aerosols, do not respond only to physical laws. While biologically based phenology functions are not readily available, current model approaches will still be based on parametric schemes strongly relying on aerobiological measurements and refined calibration procedures. In this sense, approaches like deriving





specific calibration factors per nested domain or implement 2D spatial calibration factors (e.g., Kurganskiy et al., 2020) might improve the skills of the model even in small regions. Once more reliable emission schemes will be available, the role of processes that have shown limited impact on the vertical structure of the pollen like sedimentation/dry deposition might be more relevant.

5    The proposed methodology requires a Hirst sampler nearby a polarization-sensitive lidar: first for the confirmation of the presence and the type of pollen observed, and second for the conversion of the lidar-derived backscatter coefficient into number concentration (through the retrieval of the specific extinction cross-section). To apply the methodology at sites with no Hirst sampler, but at least knowing the most probable predominant pollen type present, a look-up table would be needed. Such a table can be obtained by applying the methodology at a site with both a polarization-sensitive lidar and a Hirst sampler on a

10    large number of pollen loads with different predominant pollen types and over a relative long period of time. This is a guideline for future work.



# Appendix A. Statistics (Pearson correlation coefficient, *r*, root mean square error, *rmse*, and bias, *bias*) of the model surface pollen concentration and meteorology for 1 and 3 km.

**Table A1. Statistics (Pearson correlation coefficient *r*, root mean square error *rmse*, and *bias*) of the model surface pollen concentration daily mean and meteorology hourly mean for 1 km domain (D04) vs. measurements calculated over the period 25 March to 2 April of the event. Measurement sites detailed in Table 1.**

| XAC/XEMA site | Pollen conc. (Pollen m⁻³) | Temperature (ºC) | Wind Speed (m s⁻¹) | Wind Direction (º) | Solar Radiation (W m⁻²) | Relative Humidity (%) |
|---|---|---|---|---|---|---|
| *r* | | | | | | |
| Barcelona | 0,5 | 0,8 | 0,3 | 0,8 | 1,0 | 0,5 |
| Bellaterra/Sant Cugat | 1,0 | 0,9 | 0,2 | 0,7 | 1,0 | 0,7 |
| Girona | -0,3 | 0,9 | 0,4 | 0,5 | 0,9 | 0,8 |
| Lleida | 0,1 | 1,0 | no data | no data | no data | 0,9 |
| Manresa | -0,5 | 0,9 | 0,4 | 0,8 | 0,9 | 0,8 |
| Roquetes/Aldover | -0,3 | 0,9 | no data | no data | 0,9 | 0,9 |
| Tarragona | 0,6 | 0,8 | 0,1 | 0,3 | 1,0 | 0,6 |
| *rmse* | | | | | | |
| Barcelona | 385,4 | 1,7 | 1,8 | 70,3 | 86,6 | 15,0 |
| Bellaterra/Sant Cugat | 673,4 | 2,2 | 1,2 | 75,1 | 85,4 | 12,4 |
| Girona | 454,2 | 2,6 | 1,0 | 105,9 | 92,1 | 13,0 |
| Lleida | 102,5 | 1,6 | no data | no data | no data | 7,5 |
| Manresa | 438,3 | 2,5 | 1,2 | 89,4 | 101,2 | 15,5 |
| Roquetes/Aldover | 478,9 | 2,6 | no data | no data | 136,6 | 15,1 |
| Tarragona | 248,3 | 5,5 | 2,8 | 81,5 | 81,7 | 22,1 |
| *bias* | | | | | | |
| Barcelona | -98,6 | -0,4 | -0,4 | -0,4 | 17,2 | 6,1 |
| ellaterra/Sant Cugat | -529,2 | 0,1 | 0,4 | -11,3 | 22,4 | -1,5 |
| Girona | -324,9 | 0,3 | 0,4 | -13,5 | 8,8 | -1,4 |
| Lleida | -39,2 | -1,1 | no data | no data | no data | 2,2 |
| Manresa | 249,0 | 0,9 | 0,5 | 1,5 | 28,3 | -7,1 |
| Roquetes/Aldover | -146,2 | -2,3 | no data | no data | 37,8 | 14,2 |
| Tarragona | -111,3 | -4,8 | 0,2 | -5,4 | -31,8 | 18,6 |



**Table A2. Statistics (Pearson correlation coefficient *r*, root mean square error *rmse*, and *bias*) of the model surface pollen concentration daily mean and meteorology hourly mean for 3 km domain (D03) vs. measurements calculated over the period 25 March to 2 April of the event. Measurement sites detailed in Table 1.**

| XAC/XEMA site | Pollen conc. (Pollen m⁻³) | Temperature (ºC) | Wind Speed (m s⁻¹) | Wind Direction (º) | Solar Radiation (W m⁻²) | Relative Humidity (%) |
|---|---|---|---|---|---|---|
| *r* | | | | | | |
| Barcelona | 0,4 | 0,8 | 0,6 | 0,8 | 1,0 | 0,7 |
| Bellaterra/Sant Cugat | 1,0 | 0,9 | 0,4 | 0,8 | 1,0 | 0,8 |
| Girona | -0,3 | 0,9 | 0,4 | 0,6 | 0,9 | 0,8 |
| Lleida | 0,1 | 1,0 | no data | no data | no data | 0,9 |
| Manresa | -0,6 | 0,9 | 0,3 | 0,8 | 0,9 | 0,8 |
| Roquetes/Aldover | -0,3 | 0,9 | no data | no data | 0,9 | 0,9 |
| Tarragona | 0,6 | 1,0 | 0,5 | 0,4 | 1,0 | 0,8 |
| *rmse* | | | | | | |
| Barcelona | 458,1 | 1,8 | 1,4 | 66,6 | 79,5 | 11,8 |
| Bellaterra/Sant Cugat | 605,7 | 1,7 | 1,2 | 58,7 | 82,0 | 12,3 |
| Girona | 448,2 | 2,6 | 0,9 | 94,2 | 91,5 | 12,1 |
| Lleida | 102,6 | 1,6 | no data | no data | no data | 6,1 |
| Manresa | 953,6 | 2,4 | 1,5 | 94,0 | 103,2 | 16,5 |
| Roquetes/Aldover | 478,0 | 2,8 | no data | no data | 130,6 | 13,5 |
| Tarragona | 234,9 | 2,5 | 2,1 | 73,2 | 62,4 | 10,2 |
| *bias* | | | | | | |
| Barcelona | 130,0 | 0,0 | -0,3 | -7,6 | 17,3 | 1,2 |
| Bellaterra/Sant Cugat | -480,3 | 0,3 | 0,3 | -8,0 | 22,6 | -3,7 |
| Girona | -313,6 | 0,4 | 0,4 | -5,4 | 11,0 | -3,1 |
| Lleida | -39,5 | -1,0 | no data | no data | no data | 0,4 |
| Manresa | 665,0 | 0,9 | 0,8 | -14,7 | 32,2 | -9,2 |
| Roquetes/Aldover | -254,6 | -2,5 | no data | no data | 34,8 | 12,4 |
| Tarragona | -77,8 | -2,1 | 0,2 | 3,6 | -17,3 | 2,2 |





# Appendix B. Comparison of the model and the meteorological and pollen surface observations at XAC and XEMA sites.

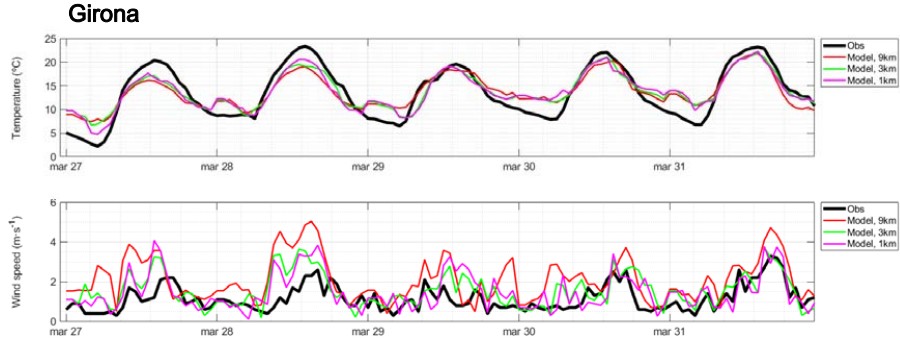

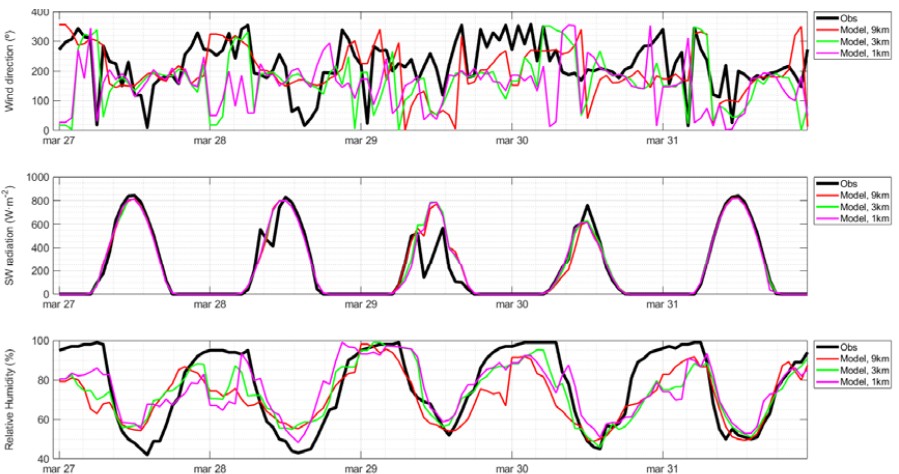

(a)





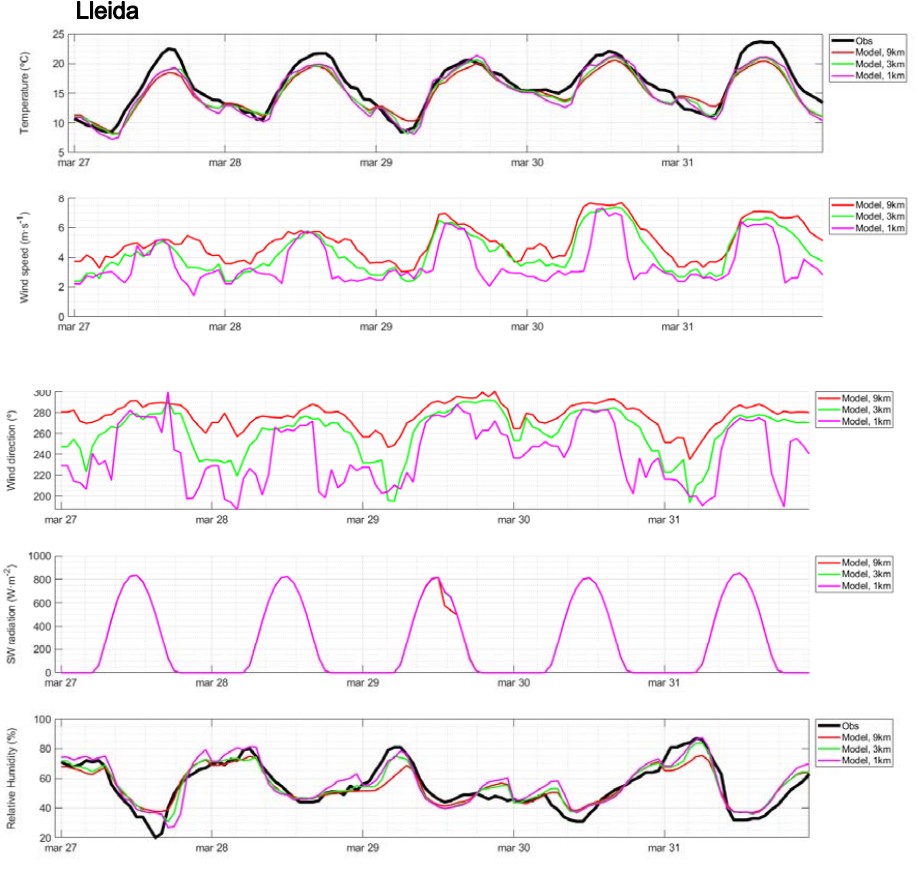

(b)



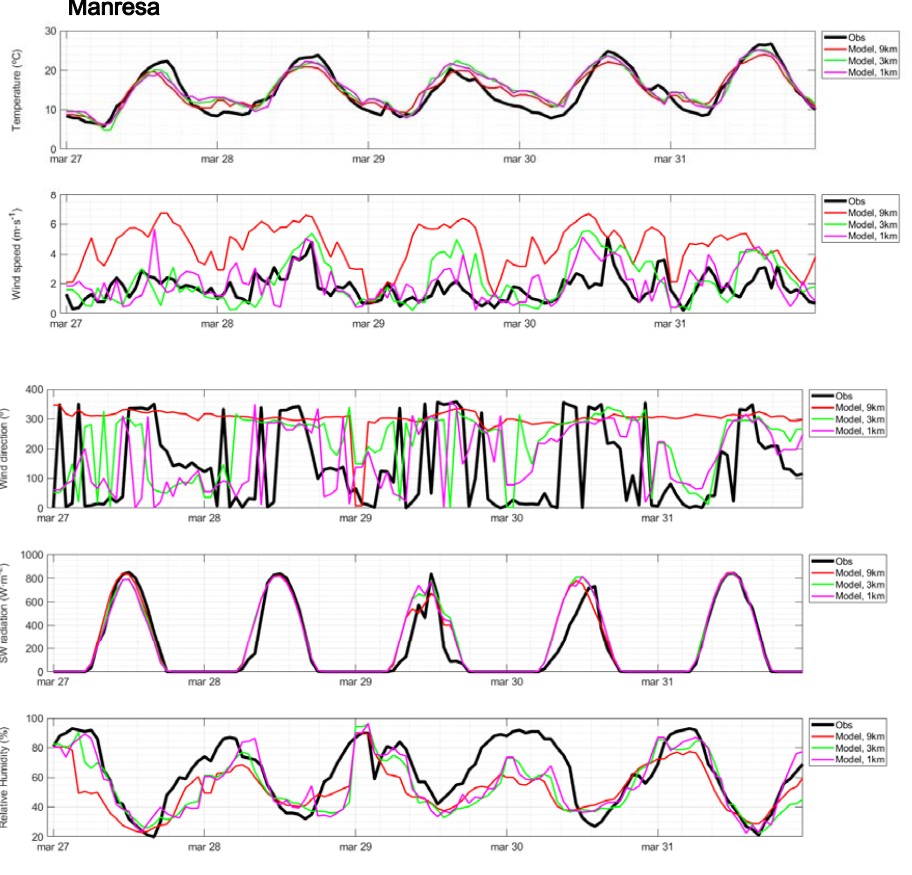

(c)



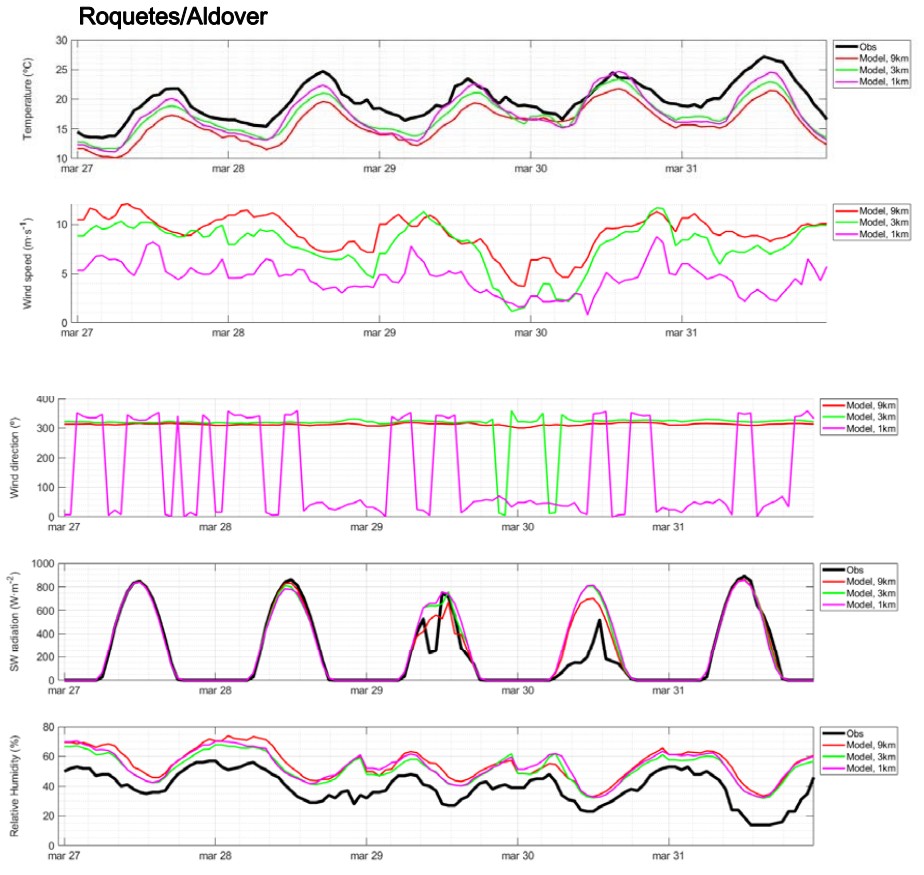

(d)



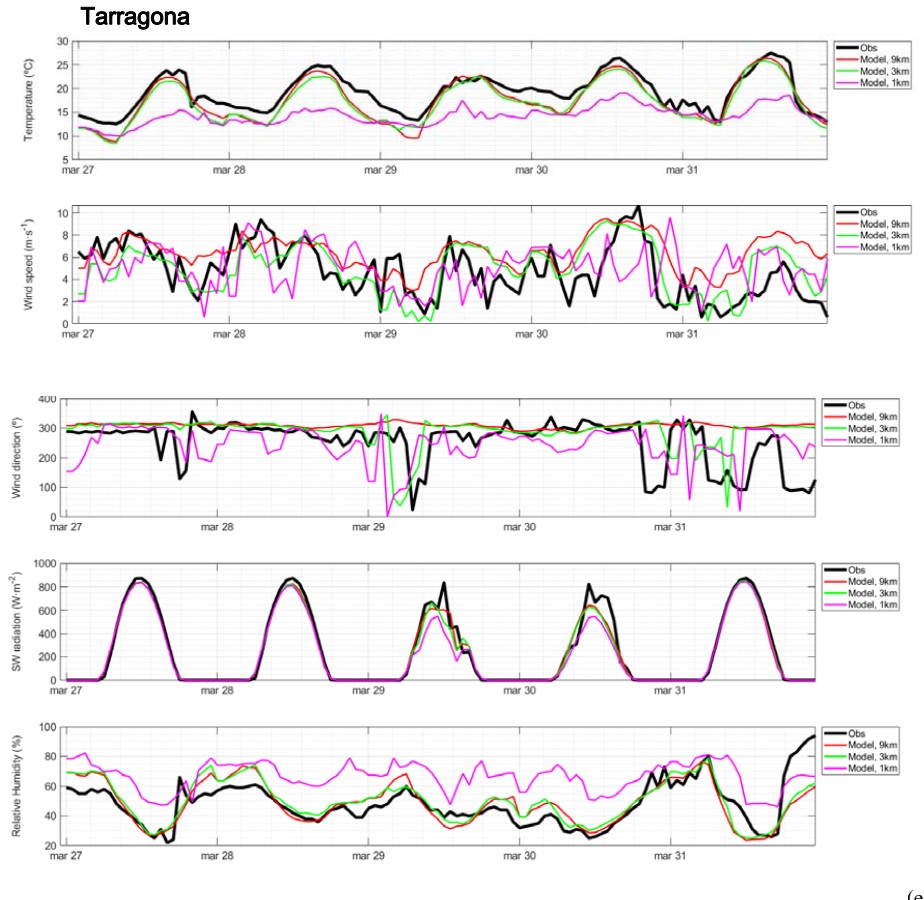

(e)

**Figure B1. Comparison of the model and the observations at (a) Girona, (b) Lleida, (c) Manresa, (d) Roquetes/Aldover, and (e)**
5 **Tarragona XEMA sites of 2-meter air temperature, 10-meter wind speed and direction, incoming shortwave radiation at surface, and 2-meter relative humidity, during the 5 days of the event. Color lines represent model results at 9, 3 and 1 km resolution.**





**Author contributions**

MS and OJ conceived the study. OJ, MA and MS designed the pollen emission scheme. OJ developed the pollen module in NMMB-MONARCH and conducted the model experiments. RI and MA provided the tree density maps. MS and AC provided the lidar measurements and MS conducted the comparison with the model. JB and CdL provided the pollen measurements and the information related to pollen, phenology and aerobiology. JJ computed the model statistics at surface and OJ conducted the comparison with the model. MS and OJ prepared the paper with contributions from all co-authors.

**Competing interests.**

The authors declare that they have no conflict of interest.

**Acknowledgments**

The authors thankfully acknowledge the computer resources at MareNostrum4 and the technical support provided by BSC (RES-AECT-2019-3-0001, RES-AECT-2020-1-0007). The authors also thank the Meteorological Service of Catalonia for providing the meteorological measurements. The MPLNET staff at NASA GSFC is warmly acknowledged for the continuous help in keeping Barcelona MPL system and the data analysis up-to-date. J. M. Baldasano is acknowledged as PI of the Barcelona MPL.

**Financial support**

Lidar data analysis were supported by funding from the H2020 program from the European Union (grant agreement no. 654109, 778349, 871115), from the Spanish Ministry of Economy, Industry and Competitiveness (ref. CGL2017-90884-REDT), from the Spanish Ministry of Science and Innovation (ref. PID2019-103886RB-I00), and the Unity of Excellence "María de Maeztu" financed by the Spanish Agencia Estatal de Investigación (ref. MDM-2016-0600). Modelling activities were supported by funding from the Ministerio de Ciencia, Innovación y Universidades as part of the BROWNING project (RTI2018-099894-BI00) and ACTRIS-España (CGL2017-90884-REDT). Airborne pollen data sampling and analyzing were supported by funding from sponsors of the Catalan Aerobiological Network (LETI Pharma, Diputació de Tarragona, Servei Meteorològic de Catalunya, Diputació de Lleida, Sociedad Española de Alergología e Inmunología Clínica (SEAIC), Societat Catalana d'Al·lèrgia i Immunologia Clínica (SCAIC), J Uriach y Compañía, S.A.) and from the Spanish Ministry of Economy, Industry and Competitiveness (ref. CGL2012-39523-C02-01, CTM2017-89565-C2-1-P and 2-P). This work is contributing to the ICTA 'Unit of Excellence' (MinECo, MDM2015-0552).

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
