# Peer review of "Measurement report: Characterization of the vertical distribution of airborne *Pinus* pollen in the atmosphere with lidar-derived profiles: a modelling case study in the region of Barcelona, NE Spain"

_Atmospheric Chemistry and Physics, 2021_

## Referee Comment (RC1)

**Review of "Measurement report: Characterization of the vertical distribution of airborne Pinus pollen in the atmosphere with lidar-derived profiles: a modelling case study in the region of Barcelona, NE Spain" by Sicard et al.**

The manuscript describes a first attempt to compare pollen concentration modelled with a 3D transport model with profiles retrieved from lidar observations. It is an interesting and timely topic as interest in bioaerosols is growing for both their impacts on health and climate. While the vertical distribution of pollen and other bioaerosols is relevant for their long-range transport and ability to impact cloud formation, it is not currently well studied. The paper is quite clearly written but could benefit from shortening. I generally support publishing the manuscript, but some revisions and clarifications are needed.

Major comments:

1. The way the references are used in the first paragraph of Introduction gives the false impression that the function and allergenicity of pollen were only recently discovered. Generally, papers that report the actual discovery should be cited, however, these facts have been known for long time and can be considered common knowledge and thus the relatively recent references to Moore and Weeb, 1983, Skjøth et al., 2013, Burczyk et al., 2004; Ellstrand, 1992; Ennos, 1994, D'Amato et al., 2007; Sofiev et al., 2017 that are attached to these statements should simply be removed.

2. Page 6, Equation (3): This equation seems to be incorrect. The authors claim to be using growth factors from Griffiths et al. (2012). Though it is not well described how they obtained the specific numbers, Griffiths et al. (2012) on their Figure 1 show average mass growth factors and the numbers presented by the authors in Table 3 look similar enough to that figure to assume it to be the source. With the assumption that the size of the pollen does not change with the water uptake as explained by the authors at the end of page 5, and given that the growth factors in Griffiths et al. (2012) are defined as the ratio of wet pollen mass to its dry mass ($\phi = \frac{m}{m_{dry}}$) the density should simply be the dry density times the growth factor:

$\rho_{pinus} = \frac{m}{V} = \frac{m_{dry} * \phi}{V} = \rho_{pinus-dry} * \phi$ , as the volume V stays constant.

3. The 8 pollen types that Griffiths et al. (2012) average to get the growth factor do not include *pinus* pollen but do show significant variability in their water uptake in higher humidity conditions. Are the simulations expected to be sensitive to this value?

4. Authors have implemented a scheme based on Helbig et al. (2004) for pollen flux to the atmosphere. Equation (4) shows the flux in grid cell x at time t to be proportional to characteristic concentration P in canopy, defined by equation (5). Eq. (5) defines P as proportional to total seasonal pollen production from all trees in the grid cell and inversely proportional to tree height and area covered by pines. While this relation does provide pollen concentration in pine canopy, in its presented form it also implies that the flux on grid cell level is inversely proportional to the area covered by pines, e.g. a dense pine grove would emit much more than sparser forest with same number of pine trees. Is this corrected for in some other part of the model not described in the paper?

5. In the paper by Helbig et al. (2004) the emission model description mentions the step of reducing the total number of available pollens by the already emitted amount. Such step is indeed necessary to keep the cumulative flux from exceeding the total pollen production. How is this implemented in the current model?

6. Page 9, line 27: "we consider the first lidar measurement (225 m) to be a proxy of what it would be at ground level" - how correct is this for 59 um particles, especially in emission regions? Comparison with the findings of recent papers by Rojo *et al* (2019) and Hugg *et al* (2020) that investigate the near ground vertical profiles of pollen concentration based on pollen traps at different heights from ground could be helpful.

7. Page 9, line 29: "In the case of the model, we consider the first model layer, the center of which varies between 24 and 24.4 m over all the simulations considered here." Is the model first layer 50 m thick also for 1 km resolution simulations? How well can a model with so thick layers represent vertical mixing in small scales?

8. Taking into account that the parameters in the formula for getting pollen concentration out of lidar observations (Eq.11) are all multiplied with each other could they be combined to a single unknown parameter (e.g. number extinction efficiency instead mass), which could simplify the presented sensitivity studies?

9. What are the assumptions behind getting *pinus* mass fraction from total pollen - what species, size and density is assumed for the other, non-*pinus* pollen and what are the uncertainties resulting from these assumptions?

10. Page 13, line 11 the authors write: " the emission source is widely distributed across the domain of interest (see Figure 1a). The latter allows a better analysis of the role of short-to-medium range transport and vertical mixing processes in the dispersion of this type of pollen." - How does the wide and uniform source distribution enhance the ability to analyze the role of transport and mixing?

11. "The model has significant skills to reproduce the temporal variability of the observations with sudden peaks on 30M and 31M or sustained high concentrations the midday of 28M or afternoon of 31M. " What is the p-value here? For instance, for daily set of 24 hourly observations, Pearson R needs to be > 0.4 to reach statistical significance at $p < 0.05$, which does not seem to be the case according to Table 7.

12. Averaging FB over whole day including also very low concentrations should be avoided as it leads to misleading results that require paragraphs of explanation. It might make more sense to compute it separately for night and day.

13. Surface emitted heavy particles are likely to produce monotonously decreasing vertical profile most of the time, so good correlation between model and observed profiles is expected and sedimentation causing this profile to uniformly shift downwards should indeed not impact the correlation much.

14. Is the daily average correlation computed hour by hour and averaged or computed over the whole daily dataset?

15. It would be informative to see the lidar derived observations of pinus pollen concentration side by side with the model simulated ones in the lidar location as a 2D time-altitude plots.

16. It's not surprising that the highest resolution model doesn't reproduce the observed features for many reasons, including imperfect wind fields and emission depending on microclimate. I would suggest comparing the variability in model and observation (standard deviations ratio, maximum and minimum values or something similar) for investigating the added value of higher resolution model.

17. Page 24, line 15: "The Hirst observations are much more variable than the model concentration and the meteorology." Hirst is a point measurement influenced by local emissions and thus it should be more variable than even 1km grid. Also, real trees are individuals that do not follow exactly the same

temporal flowering pattern and also release pollen when personally shaken by small scale turbulence. Concentrations higher up in the atmosphere are a mixture of upwind emission from many trees and thus expectedly smoother.

18. Assumption that mass fraction of pinus pollen would be constant through the whole boundary layer can be investigated using the model data by looking at the ratio of concentrations of pollens with different aerodynamic diameters simulated in the base case and enhanced runs.

19. The readability of the paper would benefit from shortening it. For instance, some detailed day by day descriptions of observations and model performance can be substantially shortened, given that the information is already present in the form of a figure or table.

Minor corrections:

1. The word "specie" is not the singular of "species", in fact, "species" is both singular and plural. Please correct throughout the manuscript.
2. Table 5 and Fig 4 duplicate contain the same data, one can be removed
3. Table 8 and Fig 11 also duplicate each other
4. Page 2, line 27: "The seminal works of Helbig et al. (2004) and Sofiev et al. (2006)" – The list of seminal works in pollen modelling should also include Schueler and Schlünzen (2006) and Schueler *et al* (2005)

**References**

Hugg T T et al 2020 The effect of sampling height on grass pollen concentrations in different urban environments in the Helsinki Metropolitan Area, Finland *PLoS One* **15** 1–12

Rojo J et al 2019 Near-ground effect of height on pollen exposure *Environ. Res.* **174** 160–9 Online: https://doi.org/10.1016/j.envres.2019.04.027

Schueler S et al 2005 Viability and sunlight sensitivity of oak pollen and its implications for pollen-mediated gene flow *Trees - Struct. Funct.* **19** 154–61

Schueler S and Schlünzen K H 2006 Modeling of oak pollen dispersal on the landscape level with a mesoscale atmospheric model *Environ. Model. Assess.* **11** 179–94

---

## Author Comment (AC1)

The manuscript describes a first attempt to compare pollen concentration modelled with a 3D transport model with profiles retrieved from lidar observations. It is an interesting and timely topic as interest in bioaerosols is growing for both their impacts on health and climate. While the vertical distribution of pollen and other bioaerosols is relevant for their long-range transport and ability to impact cloud formation, it is not currently well studied. The paper is quite clearly written but could benefit from shortening. I generally support publishing the manuscript, but some revisions and clarifications are needed.

REPLY: Thank you very much. We greatly appreciate the reviewer positive feedback. We have made a general effort to shorten the paper. In addition, Table 5 and Fig. 11 were removed. Fig. 16 was removed as well as the subsequent discussion (last paragraph before the conclusions). Appendix A and B were moved to a supplementary material. Overall the paper was reduced from 48 pages (original) to 35 (revision) pages.

Major comments:

1. The way the references are used in the first paragraph of Introduction gives the false impression that the function and allergenicity of pollen were only recently discovered. Generally, papers that report the actual discovery should be cited, however, these facts have been known for long time and can be considered common knowledge and thus the relatively recent references to Moore and Weeb, 1983, Skjøth et al., 2013, Burczyk et al., 2004; Ellstrand, 1992; Ennos, 1994, D'Amato et al., 2007; Sofiev et al., 2017 that are attached to these statements should simply be removed.

REPLY: As recommended by the reviewer we have eliminated the references related to pollen biology and its health effects.

2. Page 6, Equation (3): This equation seems to be incorrect. The authors claim to be using growth factors from Griffiths et al. (2012). Though it is not well described how they obtained the specific numbers, Griffiths et al. (2012) on their Figure 1 show average mass growth factors and the numbers presented by the authors in Table 3 look similar enough to that figure to assume it to be the source. With the assumption that the size of the pollen does not change with the water uptake as explained by the authors at the end of page 5, and given that the growth factors in Griffiths et al. (2012) are defined as the ratio of wet pollen mass to its dry mass ($\phi = \frac{m}{m_{dry}}$) the density should simply be the dry density times the growth factor:

$\rho_{pinus} = \frac{m}{V} = \frac{m_{dry} * \phi}{V} = \rho_{pinus-dry} *$ , as the volume V stays constant.

REPLY: We thank the reviewer for pointing this out as it leads to confusion and is an important comment. Equation (3) is expressed using the definition of volumetric growth factors as done for other types of atmospheric aerosols (e.g., sea salt or sulfate) in the literature (Chin et al. 2002) and not mass growth factors. This may lead to confusion as Griffiths et al. (2012) growth factors are mass based indeed and we didn't mention that the factors reported in Table 3 of the manuscript where in volume not mass. The growth factors presented in Table 3 were derived from Figure 1 of Griffiths et al. (2012). No single value is reported in Griffiths et al. (2012) per relative humidity range, and we derived a value within the range reported in the top panel of the figure representative of relative humidity range of interest and within the variability reported of growth factor values. Finally, the result was transformed to volumetric

growth factor. This approach may be confusing as we write in the text that pollen does not change their volume with changes in relative humidity. To avoid misunderstandings we've updated the manuscript using more clear formulation as suggested by the reviewer and reported the growth factors in mass and not volume. We've introduced the following changes in Section 3.1:

"Griffiths et al. (2012) describes the impact of relative humidity on the pollen density and radius. Following their results, we have implemented an increase of the Pinus pollen density with the relative humidity while the diameter of the particle is considered constant and independent of the water uptake. This effect is introduced in the model by considering relative humidity-prescribed mass-based growth factors ($\phi$=pollen_mass/pollen_mass_dry) derived from Figure 1 of Griffiths et al. (2012) shown in Table 3. Thus, the density of the particle is computed as:

$$\rho pinus = \rho pinus\_dry * \phi \qquad\qquad\qquad (3)"$$

Table 3 has been updated as follows:

"Table 3. Mass-based growth factor ($\phi$=pollen_mass/pollen_mass_dry) of *Pinus* pollen for ranges of ambient relative humidity. Values derived from Griffiths et al. (2012) Figure 1 (top panel) representative of the relative humidity range reported.

|  | Growth factor $\phi$ |
|---|---|
| 90% < Relative humidity | 1.6 |
| 80% < Relative humidity < 90% | 1.4 |
| 70% < Relative humidity < 80% | 1.2 |
| 50% < Relative humidity < 70% | 1.1 |
| RH < 50% | 1.0 |

"

Chin, M., Ginoux, P., Kinne, S., Torres, O., Holben, B. N., Dun- can, B. N., Martin, R. V., Logan, J. A., Higurashi, A., and Naka- jima, T.: Tropospheric aerosol optical thickness from the GOCART model and comparisons with satellite and sun photometer measurements, J. Atmos. Sci., 59, 461–483, 2002.

3. The 8 pollen types that Griffiths et al. (2012) average to get the growth factor do not include *pinus* pollen but do show significant variability in their water uptake in higher humidity conditions. Are the simulations expected to be sensitive to this value?

**REPLY**: Thank you for this remark. It is true that P*inus* pollen is not reported in Griffiths et al. (2012). We made the assumption that an average value reported in Griffiths et al. (2012) is a reasonable first guess of pollen taxa like the one of interest in our study.

The question raised by the reviewer about expected sensitivity in the simulations is relevant and we conducted an additional run where we decreased the growth factors 10%. The results show differences below 300 Pollen m$^{-3}$ during stable nighttime hours and negligible or below 50 Pollen m$^{-3}$ during daytime. Figure 1 shows the difference between the two simulations, the one presented in the manuscript and the one conducted decreasing the growth factors. The model is rather sensitive to the treatment of hygroscopic effects of pollen and further analysis would be required. Although this is a relevant topic, it goes beyond the scope of the present work.

[Figure]

Figure 1. Pinus number concentration (m$^{-3}$) difference between two simulations decreasing 10% the growth factors.

4. Authors have implemented a scheme based on Helbig et al. (2004) for pollen flux to the atmosphere. Equation (4) shows the flux in grid cell x at time t to be proportional to characteristic concentration P in canopy, defined by equation (5). Eq. (5) defines P as proportional to total seasonal pollen production from all trees in the grid cell and inversely proportional to tree height and area covered by pines. While this relation does provide pollen concentration in pine canopy, in its presented form it also implies that the flux on grid cell level is inversely proportional to the area covered by pines, e.g. a dense pine grove would emit much more than sparser forest with same number of pine trees. Is this corrected for in some other part of the model not described in the paper?

REPLY: Thank you for this remark. Indeed, the definition of $\gamma$ in the text is not in agreement with what we implemented in the model and described in the text following Equation 5. We do not have detailed information to quantify the number of trees per grid cell, what we did is to derive a pine tree density per grid cell over our domain of study from available cartography and density estimates per region as described in the manuscript. The tree density per grid cell is then M/$\gamma$, being $\gamma$ the area of a grid cell, not the area covered by the trees in a grid cell. This has been amended in the revised manuscript after presenting Equation (5) as follows:

"and $\gamma$ is the area  of a grid cell (m$^2$)."

5. In the paper by Helbig et al. (2004) the emission model description mentions the step of reducing the total number of available pollens by the already emitted amount. Such step is indeed necessary to keep the cumulative flux from exceeding the total pollen production. How is this implemented in the current model?

REPLY: We followed the same approach as described in Helbig et al. (2004) and other pollen models, the available pollen grains for the season is reduced by the already emitted amount of pollen grains. We missed this comment in the manuscript. We've introduced the following comment at the end of the paragraph discussing Equation (5):

"Note that the available pollen grains for the season is reduced by the already emitted amount of pollen grains at each model time step."

6. Page 9, line 27: "we consider the first lidar measurement (225 m) to be a proxy of what it would be at ground level" - how correct is this for 59 um particles, especially in emission regions? Comparison with the findings of recent papers by Rojo *et al* (2019) and Hugg *et al* (2020) that investigate the near ground vertical profiles of pollen concentration based on pollen traps at different heights from ground could be helpful.

REPLY: The Hirst collector of Barcelona is situated on the roof of a building at 23 m above ground level. This information has been added in the first paragraph of Section 4. This is in favor of our hypothesis that the first lidar measurement (at 225 m) can be used as a proxy of what it would be at the collector level (i.e. here at 23 m). Rojo et al. (2019) demonstrate that the effect of height on pollen concentrations is mainly determined by differences within the first 10 m above ground. This information has also been added to the text and the reference to Rojo et al. (2019) inserted.

7. Page 9, line 29: "In the case of the model, we consider the first model layer, the center of which varies between 24 and 24.4 m over all the simulations considered here." Is the model first layer 50 m thick also for 1 km resolution simulations? How well can a model with so thick layers represent vertical mixing in small scales?

REPLY: A first layer of 50 m is not so thick for a mesoscale model. Consider that the mid-layer height is around 25 m. Most state-of-the-art mesoscale models use similar resolution for domains from 10 to 1 km. The vertical mixing within the planetary boundary layer (PBL) strongly depends on the numerical scheme approach used in the PBL scheme of the transport model. In our case, MONARCH uses a level 2.5 local turbulence closure scheme that has been demonstrated very skillful modeling from very stable to well mixed PBL structures (Janjic, 2002; Janjic and Gall, 2012). Other schemes that fails to simulated stable conditions requires layers with lower depths to avoid the numerical diffusion of the PBL scheme used which is not the case with MONARCH model.

Janjic, Z. I.: Nonsingular implementation of the Mellor-Yamada Level 2.5 scheme in the NCEP Meso model, NCEP Office Note, No. 437, 2002.

Janjic, Z. I. and Gall, R.: Scientific documentation of the NCEP Nonhydrostatic Multiscale Model on the B grid (NMMB). Part 1 Dynamics, NCAR Technical Note NCAR/TN-489+STR, University Corporation for Atmospheric Research, 2012.

8. Taking into account that the parameters in the formula for getting pollen concentration out of lidar observations (Eq.11) are all multiplied with each other could they be combined to a single unknown parameter (e.g. number extinction efficiency instead mass), which could simplify the presented sensitivity studies?

REPLY: Yes, the conversion from backscatter coefficient to number concentration could be simplified. However we would like to keep it as is in the revised manuscript at least for two reasons. First, all parameters needed for the conversion of a lidar-derived backscatter coefficient into a pollen number concentration are there in Eq. 11, and may be useful for other scientists operating an aerosol lidar and willing to get some pollen-related information.

Second, the degree of details of Eq. 11 is necessary to perform the sensitivity analysis of Section 4.3. to see the type of dependency of $\sigma^*$ to $LR$, $D_{Pinus}$ and $\rho_{Pinus}$.

9. What are the assumptions behind getting *pinus* mass fraction from total pollen - what species, size and density is assumed for the other, non-*pinus* pollen and what are the uncertainties resulting from these assumptions?

**REPLY**: The assumption made to calculate the *Pinus* mass fraction is that the pollen mixture is made only from *Pinus* and *Platanus* taxa. For *Platanus*, the following properties were assumed (taken from Sicard et al., 2017):

Table 1 Pollen parametrization of the aerosol scheme

|  | *Pinus* | *Platanus* |
|---|---|---|
| Emission factor | 81 g/day/tree [14] | 2.48 g/day/tree [15] |
| Grain diameter | 59 μm [16] | 19 μm [7] |
| Grain density | 560 kg/m³ [16] | 920 kg/m³ [7] |

The uncertainties induced by the properties assumed for the non-*Pinus* pollen (=*Platanus*) are expected to be much smaller than the ones made for *Pinus*, as its size is 3 times smaller than that of *Pinus* which will result in a factor ~16 in terms of mass.

We also assumed that only *Pinus* and *Platanus* were present. And this is not totally true. In Sicard et al. (2016) one sees that *Cladosporium* and Cupressaceae were also present. However, in terms of number concentration, one sees from Sicard et al. (2019) that *Pinus* and *Platanus* represent more than 90 % of the number of total pollen. We neglect here the remaining ~10% composed mostly of *Cladosporium* (spores) and Cupressaceae (pollen). *Cladosporium* and Cupressaceae are also much smaller in size than *Pinus* and the neglecting of these 10% are expected to be much less in terms of volume and thus in terms of mass. Also note that the depolarizing capabilities of both taxa (*Cladosporium* and Cupressaceae) are poorly known. In case the depolarization ratios of these taxa were closer to the ones of the background aerosol than to the one of the other dominating species (*Pinus* and *Platanus*), what we retrieve as total pollen with our method would be made essentially of *Pinus* and *Platanus*.

Sicard, M., Izquierdo, R., Alarcón, M., Belmonte, J., Comerón, A. and Baldasano, J. M.: Near-surface and columnar measurements with a micro pulse lidar of atmospheric pollen in Barcelona, Spain, Atmospheric Chemistry and Physics, 16 (11), 6805-6821, doi: 10.5194/acp-16-6805-2016, 2016.

Sicard, M., Izquierdo, R., Jorba, O., Alarcón, M., Belmonte, J., Comerón, A., De Linares, C., and Baldasano, J. M.: Modelling of pollen dispersion in the atmosphere: evaluation with a continuous 1β+1δ lidar, Proc. of the 28th International Laser Radar Conference (ILRC28), EPJ Web of Conferences 176, 05006, doi: https://doi.org/10.1051/epjconf/201817605006, Bucharest (Romania), 25-30 June, 2017, 2017.

Sicard, M., Jorba, O., Izquierdo, R., Alarcón, M., De Linares, C., and Belmonte, J.: Modelling of airborne pollen dispersion in the atmosphere in the Catalonia region, Spain: model description, emission scheme and evaluation of model performance for the case of Pinus, Proc. SPIE 11152, Remote Sensing of Clouds and the Atmosphere XXIV, 111520O (9 October 2019); https://doi.org/10.1117/12.2534819, Strasbourg, France, 9 – 12 September 2019, 2019.

10. Page 13, line 11 the authors write: " the emission source is widely distributed across the domain of interest (see Figure 1a). The latter allows a better analysis of the role of short-to-medium range transport and vertical mixing processes in the dispersion of this type of pollen."

- How does the wide and uniform source distribution enhance the ability to analyze the role of transport and mixing?

**REPLY**: There are limited available in-situ measurement sites to analyze the transport of pollen across the domain of study, and only one lidar that provides information about vertical structures and mixing. An emission source widely distributed over the domain simplifies the analysis of transport processes compared with localized sources mainly because biases on the wind fields may have much more impact on the latter case when compared with the monitoring sites.

11. "The model has significant skills to reproduce the temporal variability of the observations with sudden peaks on 30M and 31M or sustained high concentrations the midday of 28M or afternoon of 31M. " What is the p-value here? For instance, for daily set of 24 hourly observations, Pearson R needs to be > 0.4 to reach statistical significance at $p < 0.05$, which does not seem to be the case according to Table 7.

**REPLY**: In this part of the discussion, this comment is qualitative and refers to Figure 9. We have added it at the beginning of the sentence ("According to Figure 9, the model …"). It is true indeed that the model overall the five days of the event is reproducing relatively well the observed concentration. In the next paragraph ("To compare the skills of the model …"), the statistical analysis of Table 6 is discussed and in the text it is clearly said that the correlation coefficient are low. "The correlations are low in most cases (around 0.2 - 0.3) highlighting the complexity to reproduce the hourly variability in pollen models.".

12. Averaging FB over whole day including also very low concentrations should be avoided as it leads to misleading results that require paragraphs of explanation. It might make more sense to compute it separately for night and day.

**REPLY**: Table 8 (now 7) has been entirely recalculated and show the diurnal means averaged over the diurnal hours from 09 to 17 UTC (following the same diurnal time period than Sicard et al., 2016). This has allowed to delete some paragraphs of explanation about the caution of analyzing daily values, averaged over periods of low concentration (night) and high concentration (day).

Sicard, M., Izquierdo, R., Alarcón, M., Belmonte, J., Comerón, A. and Baldasano, J. M.: Near-surface and columnar measurements with a micro pulse lidar of atmospheric pollen in Barcelona, Spain, Atmospheric Chemistry and Physics, 16 (11), 6805-6821, doi: 10.5194/acp-16-6805-2016, 2016.

13. Surface emitted heavy particles are likely to produce monotonously decreasing vertical profile most of the time, so good correlation between model and observed profiles is expected and sedimentation causing this profile to uniformly shift downwards should indeed not impact the correlation much.

**REPLY**: This is exactly what is observed: the correlation coefficient does not change by more than ±0.04 when deposition/sedimentation is decreased by 50%.

14. Is the daily average correlation computed hour by hour and averaged or computed over the whole daily dataset?

**REPLY**: The daily average correlation is computed hour by hour and averaged. Note that now, and following the suggestion of Comment #12, Table 8 (now 7) shows the diurnal (no more daily) FB, *r* as well as the standard deviation ratio defined diurnally, i.e. averaged from 09 to 17 UTC. In the first sentence in the paragraph below Fig. 10 the following clarification has been added "(defined as the average of the hourly values between 09 and 17 UTC)".

15. It would be informative to see the lidar derived observations of pinus pollen concentration side by side with the model simulated ones in the lidar location as a 2D time-altitude plots.

**REPLY**: All quicklooks (2D time-altitude plots) are available. As an example we show below the quicklooks of the *Basecase* simulation on 29 March at the 9 km resolution for both model and lidar in terms of backscatter coefficient. We agree with the referee that quicklooks are a nice tool to visualize qualitatively the agreement between model and measurements in the column. However for the sake of keeping the article as short as possible, we decide not to include the quicklooks in the revised manuscript, taken into account also that all the information is contained in the hourly plots of Figures 13-15 (now 12-14).

[Figure]

Model, 29M, *Basecase*, 9 km          Lidar, 29M, *Basecase*, 9 km

16. It's not surprising that the highest resolution model doesn't reproduce the observed features for many reasons, including imperfect wind fields and emission depending on microclimate. I would suggest comparing the variability in model and observation (standard deviations ratio, maximum and minimum values or something similar) for investigating the added value of higher resolution model.

**REPLY**: Following the reviewer suggestion, we have included in Table 8 (now 7) the standard deviations ratio, as well as the following discussion paragraph:

"The diurnal standard deviation ratio (**¡Error! No se encuentra el origen de la referencia.**) also illustrates the underestimation of the model during the diurnal hours on 27M and 31M ($SDR < 1$ in general) and the overestimation on 28M, 29M and 30M ($SDR > 1$ in general). On 28M, 29M and 30M, the variability of the model is larger for the 3 km resolution and generally smaller for the 9 km resolution. At this resolution the diurnal $SDR$ varies between 1.06 on 28M (variability of the model similar to the atmospheric variability) and 2.84 on 29M (variability of the model ~3 times larger than the atmospheric variability). From all the above discussion we

conclude that the 9 and 3 km resolutions are probably the most suitable to minimize bias and maximize correlation between *Pinus* pollen forecast and observations in the column. The 9 km resolution also presents the advantage of reducing the model variability. The 9 km resolution is chosen for plotting the hourly and daily mean variations shown in **¡Error! No se encuentra el origen de la referencia.**."

17. Page 24, line 15: "The Hirst observations are much more variable than the model concentration and the meteorology." Hirst is a point measurement influenced by local emissions and thus it should be more variable than even 1km grid. Also, real trees are individuals that do not follow exactly the same temporal flowering pattern and also release pollen when personally shaken by small scale turbulence. Concentrations higher up in the atmosphere are a mixture of upwind emission from many trees and thus expectedly smoother.

**REPLY**: This is right. Thank you for the clarification.

18. Assumption that mass fraction of pinus pollen would be constant through the whole boundary layer can be investigated using the model data by looking at the ratio of concentrations of pollens with different aerodynamic diameters simulated in the base case and enhanced runs.

**REPLY**: This sounds extremely interesting. The development of an altitude-dependent size model for Pinus to further perform a sensitivity analysis by comparing to the lidar vertical concentration is certainly a step necessary to go further. However we think it is at this stage clearly out of the scope of the paper. We would like the referee to notice that both *Basecase* and *Enhanced* simulations are equivalent to such an analysis. As far as mass is concerned, the parametrization of the growth factor as a function of grain density maintaining the grain volume (and thus its size) constant is equivalent to its parametrization as a function of the grain volume (and thus its size) maintaining its density constant.

19. The readability of the paper would benefit from shortening it. For instance, some detailed day by day descriptions of observations and model performance can be substantially shortened, given that the information is already present in the form of a figure or table.

**REPLY**: We have made a general effort to shorten the paper. In addition, Table 5 and Fig. 11 were removed. Fig. 16 was removed as well as the subsequent discussion (last paragraph before the conclusions). Appendix A and B were moved to a supplementary material. Overall the paper was reduced from 48 pages (original) to 35 (revision) pages.

Minor corrections:

1. The word "specie" is not the singular of "species", in fact, "species" is both singular and plural. Please correct throughout the manuscript.

2. Table 5 and Fig 4 duplicate contain the same data, one can be removed

3. Table 8 and Fig 11 also duplicate each other

4. Page 2, line 27: "The seminal works of Helbig et al. (2004) and Sofiev et al. (2006)" – The list of seminal works in pollen modelling should also include Schueler and Schlünzen (2006) and Schueler *et al* (2005)

**REPLY**: All minor comments have been corrected according to the reviewer's suggestion. #2. Table 5 and Fig. 11 have been removed.

References

Hugg T T et al 2020 The effect of sampling height on grass pollen concentrations in different urban environments in the Helsinki Metropolitan Area, Finland *PLoS One* **15** 1–12

Rojo J et al 2019 Near-ground effect of height on pollen exposure *Environ. Res.* **174** 160–9 Online: https://doi.org/10.1016/j.envres.2019.04.027

Schueler S et al 2005 Viability and sunlight sensitivity of oak pollen and its implications for pollen-mediated gene flow *Trees - Struct. Funct.* **19** 154–61

Schueler S and Schlünzen K H 2006 Modeling of oak pollen dispersal on the landscape level with a mesoscale atmospheric model *Environ. Model. Assess.* **11** 179–94

---

## Author Comment (AC2)

The manuscript presents a comprehensive, state-of-the-art study on pollen in the atmosphere (emission, vertical layering, regional transport) based on surface observation, lidar remote sensing and advanced modelling. The work includes even a detailed study on uncertainties and error sources. The paper is well written, but a bit long. Short, compact paper attract readers, too long papers often cause the opposite.

**REPLY**: Thank you very much. We greatly appreciate the reviewer positive feedback. We have made a general effort to shorten the paper. In addition, Table 5 and Fig. 11 were removed. Fig. 16 was removed as well as the subsequent discussion (last paragraph before the conclusions). Appendix A and B were moved to a supplementary material. Overall the paper was reduced from 48 pages (original) to 35 (revision) pages.

Minor revisions may further improve the paper.

**Some detailed comments:**

P3, l3: Please have also a look into the recent paper of Veselovskii, AMT, 2021 (fluorescence lidra, focus on pollen) and also Saito, Rem. Sens., 2018. Could be included in the introduction.

I miss a bit a discussion on: How did the papers of Bohlmann (ACP 2019) and Shang (ACP 2020) contributed to the field, and even improved the knowledge about pollen and lidar measurement approaches (after the pioneering papers of Noh 2013 and Sicard 2016). And what about Veselovskii, AMT 2021? I miss something like a small review (on progress) in the field of pollen and lidar applications by the expert.

**REPLY**: Bohlmann et al., 2021; Veselovskii et al., 2020; 2021 and Saito et al., 2018 have been duly added in the introduction. A sentence presenting the method developed by Shang et al. (2020) to retrieve the depolarization ratio of pure pollen (or pure pollen mixture) has been added. The recent findings of Veselovskii et al. (2020; 2021) measuring fluorescence backscattering and fluorescence capacity with broadband interference filters are now mentioned in the revised manuscript.

P3, l12-13: My request was triggered by the final, not very specific sentence of the paragraph.

**REPLY**: In this last sentence we were referring to the ongoing research made by the authors of this article, and not to the scientific community in general. The last sentence has been reformulated as follows: "The present journal paper is the apogee of the knowledge presented in the latter three conference proceedings and acquired throughout a continuous effort since 2016.".

P4: Sect. 2.2: Shang mentioned pollen lidar ratios of 65 sr.

Is your approach (methodology) is in full agreement with latest approaches (Bohlmann 2019, Shang 2020)?

Please check also Veselovskii. You will find some hints to pollen lidar ratios as well.

**REPLY**: Our methodology was also applied by Bohlmann et al. (2019) and Shang et al. (2020). However Shang et al. (2020) went a little further by presenting a new method to retrieve the

linear depolarization ratio of pure pollen (or of a mixture of pollen) present with the background aerosol from measurements of particle backscatter coefficient and depolarization ratio and Ångström exponent.

About the pollen lidar ratio at 532 nm (next all values are given at 532 nm): we used the value of 50 sr in our elastic inversions. We think it is an appropriate value for a mixture dominated by *Pinus* and *Platanus* given the large range of values found in the literature and the large variability associated to these values. In the literature, some of the values found are:

- 50±6 sr (Noh et al., 2013) for *Pinus* and *Quercus*,
- 52±12 sr (Bohlmann et al., 2019) for *Betula*,
- 62±10 sr (Bohlmann et al., 2019) for *Betula* and *Picea*,
- 61±8 sr (Shang et al., 2020) for *Betula*.
- 63±14 sr (Shang et al., 2020) for *Pinus* (scots pine).
- 40 - 60 sr (Veselovskii et al., 2021) for *Betula* and 30-60 sr for *Poaceae*.

Veselovskii et al. (2021) is not conclusive on the lidar ratio of pure pollen for both *Betula* and *Poaceae* events they observed. They say: "in many cases we observed lidar ratios below 40 sr at both wavelengths. However, we also had cases when the lidar ratios at both wavelengths were in the 50–60 sr range. Thus, at the moment we are not able to specify lidar ratios for pure pollen". One sees that within the same taxon, the interval of values, taking into account the error bars, can be quite large. For *Betula* it varies between 40 and 72 sr. For *Pinus* it varies between 44 and 77 sr.

Concerning depolarization ratios: What about masking effects? Dust or dry-marine-related depolarization enhancements? I think, these effects are negligible. But there are several field sites (cities) close to the Mediterranean Sea (and there will be sea breeze effects, advecting sea salt particles across the coastal regions…, sometimes up to 10-20 km into the continent). Please, provide a short comment on this.

**REPLY**: For sure dust was not present during the five days of measurements presented. This has been checked already in Sicard et al. (2016) with the dust transport models BSC-DREAM8b v2 (Barcelona Supercomputing Center – Dust Regional Atmospheric Model 8 bins) and NMMB/BSC-DUST (Nonhydrostatic Multiscale Meteorological Model on the B grid/Barcelona Supercomputing Center – Dust), as well as HYSPLIT (Hybrid Single Particle Lagrangian Integrated Trajectory) back trajectories.

As far as dry marine aerosols are concerned, Barcelona is a coastal city and sea breeze effects are present. During the event, the relative humidity at ground level during daytime varied between 40 and 70 % (Fig. 3 of Sicard et al., 2016). And the tendency on all days was to increase with increasing height (see figure below from radiosounding measurements performed daily at 12 UTC). This behavior is the opposite of Haarig et al. (2017) observations of dry marine particles. According to Haarig et al. (2017), values of depolarization ratio of ~0.15 can be reached for dry marine particles. This occurs in the top part of the boundary layer when the relative humidity falls to values on the order of 40%. Given the above, it is highly unlikely that dry marine particles have been present in the measurements presented in the paper. For information, the UPC lidar team, which develops and operates aerosol lidars since the late nineties has never experienced the presence of depolarizing, dry marine particles in the PBL.

[Figure]

Haarig, M., Ansmann, A., Gasteiger, J., Kandler, K., Althausen, D., Baars, H., Radenz, M., and Farrell, D. A.: Dry versus wet marine particle optical properties: RH dependence of depolarization ratio, backscatter, and extinction from multiwavelength lidar measurements during SALTRACE, Atmos. Chem. Phys., 17, 14199–14217, https://doi.org/10.5194/acp-17-14199-2017, 2017.

P7, Figure 1 is very nice and rather helpful. If possible, provide a bar, indicating 10, 20 or 50 km distance…, in the right lower corner… In Figure 1, one can see that sea breeze (and land breeze effects in the night) will affect the pollen transport.

**REPLY**: Thank you for your comment. We've included a length bar in the right lower corner of the image following the reviewer suggestion. We agree with the reviewer comment that the dispersion in the region is significantly dominated by land-sea breeze circulations, especially during summertime. However, the period of study was not characterized by this type of circulation.

P12, Figure 3b is very convincing, shows excellent agreement! I did not expect such an agreement between a point observation (at ground) and a column observation (lidar). This corroborates that dust and dry sea salt effects are probably negligible. Should be mentioned.

**REPLY**: True. It also validates our hypothesis that the first lidar measurement (225 m) is a good proxy of what it would be at ground level.

P15, Figure 6: Please mention that the solid line (in each plot) shows the coast line (and not a river…, as I was thinking in the first moment).

**REPLY**: This explanation has been added in the caption of Fig. 6.

P24, Figure 16 is mentioned… To my opinion, the lidar profiles are biased (above 400m) . The profiles are unrealistically smooth. I speculate that the pure Rayleigh depolarization background is varying with time. By using a fixed, but a bit smaller values than the actual

(instrument-related) background Rayleigh value, you get such a bias. But in reality, it is the background, and not a pollen depolarization contribution. At least it looks strange.

**REPLY**: The pure Rayleigh depolarization background may vary with time if the temperature varies. This has been studied in the past by Behrendt et al. (2002). The following figure taken from that paper shows how the Rayleigh depolarization ratio varies with temperature for different bandwidths of the interference filter used at 532 nm. One sees that the absolute values are small (<0.015) and that the variations with temperature are also small. For information the bandwidth of the interference filter of our MPL is smaller than 0.2nm.

Anyhow Fig. 16 and the discussion (last paragraph before the conclusions) have been deleted in the revised manuscript. Referee #1 suggested to avoid to include the nighttime profiles in the calculation of the statistics in Table 8. This allowed us to delete the lengthy discussion about possible nighttime residual layers observed by the lidars and underestimated by the models.

[Figure]

Fig. 4. Molecular volume depolarization ratio $\delta_{mol}$ against temperature $T$ for different values of the width of the transmission band of the lidar receiver $FWHM$ calculated for Gaussian-shape transmission bands centered at a laser wavelength of 532 nm.

Behrendt, A. and Nakamura, T.: Calculation of the calibration constant of polarization lidar and its dependency on atmospheric temperature, Optics Express, 10, 805–817, 2002.

Appendix B: Comparison of meteorological and pollen surface observations with respective model results…… sounds better… However, do we need Appendix B?

**REPLY**: Appendix A and B have been moved to Supplementary materials.

---

## Author Response (AR2)

We warmly thank again the editor for this final review of our paper and of our answers to the referee's comments.

In particular, please address the following points in your revisions:
1. The author responses to Reviewer 1, comments 9 & 10 contain important information, which should be incorporated into the manuscript (in summary form) so that it is available to readers.

**REPLY**: The following sentence, page 11, line 15-20, has been added:

"The main assumption made to calculate the *Pinus* mass fraction is that the pollen mixture is assumed to be made only from *Pinus* and *Platanus* taxa, whereas *Cladosporium* and Cupressaceae were also present (Sicard et al., 2016a). However, in terms of number concentration, one sees from Sicard et al. (2019) that *Pinus* and *Platanus* represent more than 90 % of the number of total pollen. Our assumption is equivalent to neglecting the remaining ~10 %. As *Cladosporium* and Cupressaceae are much smaller in size than *Pinus*, our assumption is equivalent to neglecting in terms of mass probably less than 1 % of the total mass."

2. With respect to Reviewer 1, comment 11 - a model should be termed skillful only if it has been shown to outperform a reference model according to objective metrics. Please revise accordingly.

**REPLY**: There is no *Pinus* reference model to which our results can be compared. The word "skill" has been deleted in the text and we now talk about model performances.